# Adversarially Robust Control of Conditional Value-at-Risk Via Rockafellar-Uryasev Conformal Inference

Catherine Chen [1]  Jingyan Shen [2]  Zhun Deng [3]  Lihua Lei [4]

## Abstract

We present an online, distribution-free framework for controlling the Conditional Value-at-Risk (CVaR), extending conformal tail risk control to non-stationary and adversarial environments. Unlike classical risk control methods, which rely on stationarity or linearity of expectation, our approach provides provable safety guarantees for a *nonlinear tail risk functional* under *arbitrary* data-generating processes that may *drift or shift strategically over time*. By leveraging deep connections between conformal tail risk control, online learning, and the variational representation of CVaR introduced by Rockafellar & Uryasev (2000), we develop a novel procedure for online CVaR control with adversarial regret guarantees. The proposed method operates without assumptions on the underlying data-generating process, making it broadly applicable in modern high-stakes deployment settings. We prove that the realized empirical CVaR is asymptotically controlled at the target level, and that the resulting control is asymptotically tight up to a finite-sample $O(1/\sqrt{T})$ conservatism gap. We demonstrate the effectiveness of our approach on portfolio risk management and toxicity mitigation for Large Language Models (LLMs), where rare but catastrophic failures dominate system risk.

## 1. Introduction

Modern machine learning systems are increasingly deployed in high-stakes and safety-critical settings, including financial decision-making, automated content moderation, and

[1]Institute for Computational and Mathematical Engineering, Stanford University [2]Department of Computer Science, New York University [3]Department of Computer Science, University of North Carolina at Chapel Hill [4]School of Business and Department of Statistics (by courtesy), Stanford University. Correspondence to: Catherine Chen <cyc2152@stanford.edu>.

*Proceedings of the 43rd International Conference on Machine Learning*, Seoul, South Korea. PMLR 306, 2026. Copyright 2026 by the author(s).

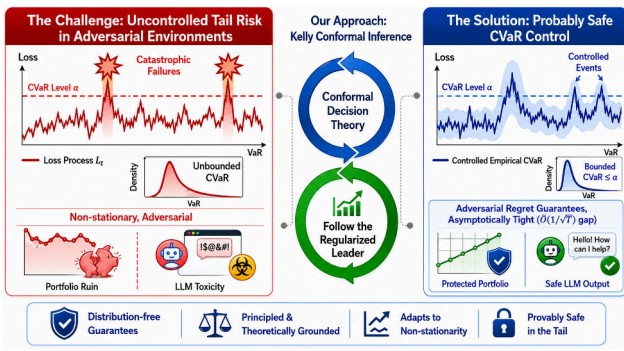

*Figure 1.* **Online** CVaR **control via Rockafellar-Uryasev Conformal inference.** Left: Uncontrolled tail risk in adversarial or non-stationary environments leads to catastrophic failures. Middle: Our method combines Conformal Decision Theory with AdaGrad-FTRL. Right: The resulting controller provably enforces the target CVaR level with adversarial guarantees.

large language model (LLM) alignment. In such applications, controlling the average risk is not enough to ensure system reliability, where a single catastrophic error can dominate operational, legal, or reputational risk. This has led to growing interest in *tail risk control* (Snell et al., 2022; Zollo et al., 2024b; Chen et al., 2025), where the goal is to design learning and decision-making systems whose worst-case outcomes remain within acceptable limits.

Conditional Value-at-Risk ($\text{CVaR}_\beta$) is a canonical and widely adopted measure of tail risk, capturing the expected loss conditioned on the worst $100(1-\beta)\%$ of outcomes (Rockafellar & Uryasev, 2000). Unlike average risk measures, CVaR explicitly focuses on rare but catastrophic events, making it suitable for high-stakes, adversarial, and non-stationary environments in which failures could be costly and strategically induced. For instance, in finance, CVaR is a fundamental concept in portfolio optimization and regulatory risk management.

In modern machine learning systems, an analogous notion arises naturally. For instance, in LLM deployment, CVaR measures the expected severity of the most harmful generations. Critically, many deployment environments for LLMs are inherently *non-stationary*. Therefore, the ability to control CVaR in an *online* manner is essential. Overall,

$\text{CVaR}_\beta$ provides a principled objective for stress testing and red-teaming, as demonstrated in existing work on adversarial prompting and automated test-case generation for uncovering model vulnerabilities (Perez et al., 2022).

In this work, we address whether $\text{CVaR}_\beta$ can be controlled in an *online* and *adversarial* environment. Unlike expected loss, $\text{CVaR}_\beta$ is a nonlinear, tail-sensitive risk functional that depends on the extreme quantiles of the loss distribution. Guarding against the rare but catastrophic outcomes renders tail risk control fundamentally costly, as the events that characterize $\text{CVaR}_\beta$ are precisely those that are least frequently observed.

To date, existing online risk control and conformal calibration methods fail to tackle the intrinsic nonlinearity of $\text{CVaR}_\beta$ (Gibbs & Candes, 2021; Yang et al., 2024; Lekeufack et al., 2024). In particular, widely used procedures such as Adaptive Conformal Inference (ACI) (Gibbs & Candes, 2021), Bellman Conformal Inference (BCI) (Yang et al., 2024), and Conformal Decision Theory (CDT) (Lekeufack et al., 2024) fundamentally rely on linearity of expectation and therefore break down for $\text{CVaR}_\beta$. While recent conformal tail risk control methods can provide statistical guarantees for $\text{CVaR}_\beta$ via L-statistics and uniform bounds (Chen et al., 2025; Snell et al., 2022; Deng et al., 2023; Zollo et al., 2024a; Deng et al., 2025), these guarantees do not extend to online, non-stationary, or adversarial regimes. Overcoming these limitations requires new techniques that explicitly exploit the variational structure of $\text{CVaR}_\beta$.

In this work, we develop an online, distribution-free framework for tail risk control that provides provable safety guarantees under non-stationary and adversarial data. Our approach builds on the Rockafellar-Uryasev (RU) variational representation of CVaR (Rockafellar & Uryasev, 2000) and reduces online tail risk control to a two-level online optimization problem, combining Conformal Decision Theory (Lekeufack et al., 2024) with AdaGrad-Follow the Regularized Leader (AdaGrad-FTRL) (McMahan, 2011). The resulting procedure requires no distributional assumptions, and provably controls the realized empirical $\text{CVaR}$ at the target level, with a vanishing tightness gap.

**Contributions.** We summarize our main contributions as follows:

- We formulate the problem of *online* $\text{CVaR}$ *control* under non-stationary and adversarial data, extending existing conformal and risk calibration methods that rely on stationarity or linearity of expectation.

- We show that the RU variational representation enables a principled reduction of nonlinear tail risk control to an online regret minimization problem over an auxiliary threshold parameter.

- We provide a *provable safety guarantee*: the empirical $\text{CVaR}$ of the realized sequence is asymptotically controlled at the target level, even under adversarial data. Moreover, the resulting control is asymptotically tight up to a finite-sample $O(1/\sqrt{T})$ conservatism gap.

- We demonstrate the practical effectiveness of our method on LLM toxicity control and portfolio management under distribution shift.

### 1.1. Related Work

**Rockafellar-Uryasev (RU) Representation.** In Rockafellar & Uryasev (2000), Rockafellar and Uryasev introduce a variational representation of $\text{CVaR}_\beta$ that has become the foundation for risk-sensitive optimization, transforming a tail risk objective into a tractable optimization problem over an auxiliary threshold parameter. Several recent works have highlighted the central role of this variational form in offline robust decision-making under distribution shift and data bias (Sahoo et al., 2025; Lei et al., 2023). In contrast, our setting is fundamentally online as data arrives sequentially, distributions may drift, or be chosen adaptively, and the goal is not merely to optimize a static $\text{CVaR}$ objective, but to dynamically control tail risk over time. Our results therefore complement and extend this line of work by bringing $\text{CVaR}$-based risk control into the sequential, adversarial regime.

**Conformal Decision Theory.** In Conformal Decision Theory (CDT) (Lekeufack et al., 2024), a single calibration parameter suffices to control the expected loss of a broad class of decisions, provided a suitable "safeguard" action is available. Similar to conformal risk control (Bates et al., 2021; Angelopoulos et al., 2025a;b), their technique does not work for nonlinear risk functionals like CVaR. In contrast, our work exploits a fundamentally different strategy than CDT.

**Gradient Equilibrium.** Angelopoulos et al. (2025c) argue that the standard notion of regret is not directly relevant to online uncertainty quantification or risk control problems and propose gradient equilibrium as an alternative. By contrast, in our setting, while regret is not the final objective, it plays a crucial role. We demonstrate that by minimizing a carefully chosen regret objective, one can provably control $\text{CVaR}_\beta$ at a desired level in adversarial and non-stationary environments. This deepens the connection between online learning and online risk control.

**Adaptive FTRL (AdaGrad–FTRL).** A central challenge presented by CVaR control is a secondary online convex optimization problem over the RU threshold variable ($c_t$). While many no-regret online optimization algorithms are available, we seek a method that is stable, compatible with constrained continuous action variables, and does not re-

quire manual tuning of a learning rate for the threshold update. To this end, we adopt an adaptive Follow the Regularized Leader (FTRL) update with AdaGrad-style scaling (McMahan, 2011; Shalev-Shwartz, 2012; Duchi et al., 2011). At each round, the threshold variable is selected by minimizing the cumulative surrogate loss with a quadratic regularization. This adaptive scaling automatically adjusts the effective step size based on observed loss behavior, yielding a stable and self-tuning update rule while preserving standard no-regret guarantees.

## 2. Rockafellar-Uryasev Conformal Inference

We present our two-level online learning procedure for controlling $\mathrm{CVaR}_\beta$. Section 2.1 discusses the online setting in which our algorithm operates. In Section 2.2, we develop the connection between online CVaR control and online regret minimization for an extended game by leveraging the RU representation (Rockafellar & Uryasev, 2000). We discuss the AdaGrad-FTRL method in Section 2.3, followed by the main theoretical guarantee. In Sections 2.4 and 2.5, we analyze the outer and inner updates of the algorithm and discuss their theoretical properties.

### 2.1. Setting

We consider an online setting indexed by time $t = 1, 2, \ldots, T$. At each time point $t$, the decision-maker chooses a calibration parameter $\lambda_t$ and Nature follows by choosing a loss function $R_t(\lambda)$. We assume that all functions $R_t$ map $[\lambda_{\min}, \lambda_{\max}]$ to $[R_{\min}, R_{\max}]$ for some $-\infty < \lambda_{\min} < \lambda_{\max} < \infty$ and $-\infty < R_{\min} < R_{\max} < \infty$. The decision-maker's choice can rely on $(R_{t-1}(\cdot), \lambda_{t-1}, R_{t-2}(\cdot), \lambda_{t-2}, \ldots)$ and Nature's choice can depend on the decision-maker's information set at time $t$ together with $\lambda_t$. The realized loss for the decision-maker occurred at the end of time $t$ is given by $R_t(\lambda_t)$, where we extend the domain of $R_t(\lambda_t)$ In Section 3 to the entire real line with

$$R_t(\lambda) = \begin{cases} R_{\min}, & \lambda < \lambda_{\min}, \\ R_{\max}, & \lambda > \lambda_{\max}. \end{cases}$$

we consider two applications in portfolio management and toxicity control for LLMs. In both examples, $\lambda_t$ is a scalar in $[0, 1]$ and thus the decision-maker just needs to choose a scalar in each round.

### 2.2. Bounding CVaR by Regret

Unlike expectation, $\mathrm{CVaR}_\beta$ is a *nonlinear* functional of the loss distribution and cannot be written as a simple average of per-round losses. As a consequence, classical notions of regret based on summing per-round losses do not apply directly.

Given a sequence of realized losses $R_1, \ldots, R_T$, we define the empirical CVaR through the Rockafellar–Uryasev variational representation:

$$\widehat{\mathrm{CVaR}}_\beta(R_{1:T}) := \min_{c \in \mathbb{R}} \left\{ c + \frac{1}{1-\beta} \cdot \frac{1}{T} \sum_{t=1}^{T} (R_t - c)_+ \right\}.$$

Since the losses are bounded in $[R_{\min}, R_{\max}]$, the minimization may equivalently be restricted to $c \in [R_{\min}, R_{\max}]$.

In the absence of ties, $\widehat{\mathrm{CVaR}}_\beta(R_{1:T})$ can be approximated by the average of losses among the top $100(1-\beta)\%$ of losses, i.e.,

$$\widehat{\mathrm{CVaR}}_\beta(R_{1:T}) = \frac{1}{\lfloor T(1-\beta) \rfloor} \sum_{i > \lceil T\beta \rceil} R_{(i)} + O\left(\frac{1}{T}\right)$$

where $R_{(1)} \leq R_{(2)} \leq \ldots \leq R_{(T)}$ are the ordered statistics. The remainder term $O(1/T)$ goes away when $T\beta$ is an integer.

With the definition of empirical CVaR, we say the decision-maker achieves an online CVaR control at some target level $\alpha \in (R_{\min}, R_{\max})$ iff

$$\widehat{\mathrm{CVaR}}_\beta(R_1(\lambda_1), \ldots, R_T(\lambda_T)) \leq \alpha + o(1). \quad (1)$$

In contrast to standard cumulative loss, $\widehat{\mathrm{CVaR}}_\beta(R_{1:T})$ depends on the entire loss sequence in a non-additive manner. In particular, changing a single large loss can alter both the quantile threshold and the set of points entering the tail average. Using the Rockafellar-Uryasev representation of $\mathrm{CVaR}_\beta$ (Rockafellar & Uryasev, 2000) we can reformulate the criterion (1) as follows:

$$\min_{c \in [R_{\min}, R_{\max}]} \frac{1}{T} \sum_{t=1}^{T} \left[ c + \frac{1}{1-\beta}(R_t - c)_+ \right] \leq \alpha$$

This representation suggests an extended game between the decision-maker and Nature. At time $t$, the decision-maker chooses an auxiliary variable $c_t$ in addition to $\lambda_t$ in each round $t$ and Nature chooses $R_t(\lambda)$ as before. The cumulative realized loss is given by

$$\frac{1}{T} \sum_{t=1}^{T} \left[ c_t + \frac{1}{1-\beta}(R_t(\lambda_t) - c_t)_+ \right].$$

For this extended game, we can define the regret as usual:

$$\mathrm{Reg}_T(c) = \sum_{t=1}^{T} \left[ c_t + \frac{1}{1-\beta}(R_t(\lambda_t) - c_t)_+ \right]$$
$$- \sum_{t=1}^{T} \left[ c + \frac{1}{1-\beta}(R_t(\lambda_t) - c)_+ \right]. \quad (2)$$

This regret definition is consistent with the typical objective of no-regret online learning optimization, where the first term tracks the average loss accumulated by the controller, and the second term is the best fixed action in hindsight.

With this regret, we can turn the CVaR control problem into an expected risk control problem.

**Proposition 2.1.** *If the decision-maker can choose $(\lambda_t, c_t)$ such that*

$$\left| \max_{c \in [R_{\min}, R_{\max}]} \operatorname{Reg}_T(c) \right| = o(T), \qquad (3)$$

*and*

$$\frac{1}{T} \sum_{t=1}^{T} \left[ c_t + \frac{1}{1-\beta}(R_t(\lambda_t) - c_t)_+ \right] \leq \alpha + o(1), \quad (4)$$

*then* (1) *holds.*

Proposition 2.1 implies that it remains to achieve (3) and (4). In the following, we will apply an AdaGrad-FTRL algorithm for (3) and a CDT-style algorithm to achieve (4). Notably, (3) requires bounding the regret $\max_c \operatorname{Reg}_T(c)$ from both above and below. As we will see in the proofs, the upper bound is crucial to prove (4) while the lower bound is used to conclude CVaR control from (4).

## 2.3. Rockafellar–Uryasev Conformal Inference and Regret Bound

Our algorithm, shown in Algorithm 1, has a two-level optimization structure. The outer level performs updates on a control parameter $\lambda_t$ to enforce the $\mathrm{CVaR}_\beta$ constraint using CDT (or ACI/BCI-style) update (Lekeufack et al., 2024; Gibbs & Candes, 2021; Yang et al., 2024). Meanwhile, the inner level solves a one-dimensional convex optimization problem induced by the Rockafellar–Uryasev representation, in an online manner. Using AdaGrad-FTRL updates (McMahan, 2011; Duchi et al., 2011), the inner level adaptively learns the optimal quantile threshold $c_t$, yielding adversarial regret guarantees without step-size tuning.

We first present our main regret bound in Theorem 2.2.

**Theorem 2.2.** *Assume the loss $R_t(\lambda)$ is uniformly bounded for all $t$ and $\lambda$. Then the proposed algorithm guarantees*

$$\widehat{\mathrm{CVaR}}_\beta(R_{1:T}) \ \leq \ \alpha + CT^{-1/2},$$

*for some constant $C$ that only depends on $\beta$, $\alpha$, $\gamma$, $\lambda_{\min}$, $\lambda_{\max}$, $R_{\min}$, $R_{\max}$.*

*Remark* 2.3 (Price of nonlinearity). In classical online control of expected loss, analogous guarantees typically achieve a $O(1/T)$ convergence rate due to the linearity of the objective (Lekeufack et al., 2024). In contrast, $\mathrm{CVaR}_\beta$ is a nonlinear functional that depends on the extreme tail of the

---

**Algorithm 1** RU Conformal Inference

**Input:** CVaR level $\beta$, target risk level $\alpha$, outer relative step size $\gamma_0$, effective domain $[\lambda_{\min}, \lambda_{\max}]$, range $[R_{\min}, R_{\max}]$, initial $\lambda_1$.

**Procedure:**
$c_1 \leftarrow (R_{\min} + R_{\max})/2$.
$q_0 \leftarrow \max\{1, \beta/(1-\beta)\}^2$.
$\gamma \leftarrow \gamma_0(\lambda_{\max} - \lambda_{\min})$.
**for** $t = 1, \dots, T$ **do**

  **Nature move:**
  Observe loss function $R_t(\cdot)$

  **Extend the loss function:**
  Extend $R_t$ with $R_t(\lambda) \leftarrow \begin{cases} R_{\min}, & \lambda < \lambda_{\min}, \\ R_{\max}, & \lambda > \lambda_{\max}. \end{cases}$

  **Construct RU surrogate:**
  $\ell_t^{RU} \leftarrow c_t + \frac{1}{1-\beta}(R_t(\lambda_t) - c_t)_+$.

  **Outer-level CDT update:** $\lambda_{t+1} \leftarrow \lambda_t - \gamma(\ell_t^{RU} - \alpha)$.

  **Inner-level AdaGrad–FTRL update:**

  $g_t \leftarrow 1 - \frac{1}{1-\beta} \cdot \mathbf{1}\{R_t(\lambda_t) > c_t\}$.

  $q_t \leftarrow q_{t-1} + g_t^2, \qquad \eta_t \leftarrow \frac{1}{2\sqrt{q_t}}$.

  $c_{t+1} \leftarrow \underset{c \in [R_{\min}, R_{\max}]}{\arg\min} \left\{ \frac{1}{2\eta_t}\left(c - \frac{R_{\min} + R_{\max}}{2}\right)^2 \right.$

  $\left. + \sum_{s=1}^{t}\left(c + \frac{1}{1-\beta}(R_s(\lambda_s) - c)_+\right) \right\}$.

**end for**

---

loss distribution. As a result, our guarantee incurs a slower $O(1/\sqrt{T})$ rate, reflecting the statistical and algorithmic cost of learning and controlling tail risk from rare events.

In the next two subsections, we walk through the proof of Theorem 2.2 by studying the theoretical properties of the inner- and outer-level updates.

## 2.4. Achieving (3) Via Inner-level Update

In this and next subsections, to simplify exposition, we assume $\lambda_{\min} = R_{\min} = 0$ and $\lambda_{\max} = R_{\max} = 1$. In general, this can be achieved by replacing $\alpha, R(\lambda)$ by $\tilde{\alpha}, \tilde{R}(\tilde{\lambda})$ where

$$\tilde{\alpha} = \frac{\alpha - R_{\min}}{R_{\max} - R_{\min}}, \tilde{\lambda} = \frac{\lambda - \lambda_{\min}}{\lambda_{\max} - \lambda_{\min}}, \qquad (5)$$

and

$$\tilde{R}(\tilde{\lambda}) = \frac{R(\lambda_{\min} + (\lambda_{\max} - \lambda_{\min})\tilde{\lambda}) - R_{\min}}{R_{\max} - R_{\min}}. \qquad (6)$$

Fixing the sequence $\lambda_1, \lambda_2, \ldots$, the incremental loss function in the extended game is expressed as

$$f_t(c) = c + \frac{1}{1-\beta}\big(R_t(\lambda_t) - c\big)_+, \qquad (7)$$

which is convex and $G$-Lipschitz on $[0,1]$, where one valid subgradient of $f_t$ at $c$ is

$$g_t(c) \in \partial f_t(c) = 1 - \frac{1}{1-\beta}\mathbf{1}\{R_t(\lambda_t) > c\},$$

and hence

$$g_t(c) \in \left\{1, -\frac{\beta}{1-\beta}\right\}, \qquad G = \max\left\{1, \frac{\beta}{1-\beta}\right\}.$$

Our goal is to choose $c_1, \ldots, c_T$ in an online manner to minimize the regret with respect to a fixed single decision $c$:

$$\mathrm{Reg}_T(c) = \sum_{t=1}^{T} f_t(c_t) - \sum_{t=1}^{T} f_t(c).$$

A major practical challenge in online learning is that the optimal learning rate can vary dramatically across regimes. In benign or stationary environments, aggressive updates can significantly accelerate convergence. Conversely, in noisy, heavy-tailed, or adversarial settings, large step sizes can lead to instability and erratic behavior.

To avoid tuning a learning rate for $c_t$ while maintaining adversarial guarantees, we use an AdaGrad-FTRL algorithm. Let $g_t$ be a chosen subgradient at the current iterate. The update rule of AdaGrad-FTRL then reduces to

$$c_{t+1} \leftarrow \arg\min_{c\in[0,1]} \frac{1}{2\eta_t}(c - 1/2)^2$$
$$+ \sum_{s=1}^{t}\left\{c + \frac{1}{1-\beta}(R_s(\lambda_s) - c)_+\right\},$$

where the adaptive step size is set to be

$$\eta_t = \frac{1}{2\sqrt{q_t}}, \quad q_t = q_{t-1} + g_t^2, \quad q_0 = \max\left\{1, \frac{\beta}{1-\beta}\right\}^2.$$

**Theorem 2.4.** *For every adaptive or adversarial sequence $R_1(\cdot), \ldots, R_T(\cdot) : [0,1] \mapsto [0,1]$, the iterates above satisfy*

$$-\frac{1}{4}\sqrt{q_T} \le \max_{c\in[0,1]} \mathrm{Reg}_T(c) \le \frac{3}{4}\sqrt{q_T}.$$

*In particular,*

$$\left|\max_{c\in[0,1]} \mathrm{Reg}_T(c)\right| \le \frac{3}{4}\max\left\{1, \frac{\beta}{1-\beta}\right\}\sqrt{T+1}.$$

We can show that the leading term is optimal in terms of the rate in $T$ and the constant is looser by a constant factor. The proof is presented in Appendix B.4.

**Theorem 2.5.** *For every $T \ge 1$ and every online learning algorithm, possibly randomized, there exists a sequence $R_1(\cdot), \ldots, R_T(\cdot) : [0,1] \mapsto [0,1]$, such that*

$$\mathbb{E}\left[\max_{c\in[0,1]} \mathrm{Reg}_T(c)\right] \ge \frac{1}{\sqrt{2\pi}}\sqrt{\frac{\beta}{1-\beta}}\sqrt{T} \cdot (1 + o(1)),$$

*where the expectation is only over the algorithmic randomization. For deterministic algorithms, the expectation may be omitted.*

When $\beta > 1/2$, the constant gap can be closed by allowing the tuning parameter to depend on $T$. We present the algorithm and its regret upper bound in Appendix B.5. Nevertheless, we prefer the AdaGrad-FTRL algorithm in practice that is agnostic to the horizon $T$.

## 2.5. Achieving (4) Via Outer-level Updates

Fixing the sequence $c_1, c_2, \ldots$, (4) becomes the same objective for CDT with additive cumulative losses. When $R_t(\lambda)$ is bounded, we can prove the risk control up to an $O(1/\sqrt{T})$ factor. The proof is substantially more complicated than the argument in Lekeufack et al. (2024) because the sequence $\{c_t\}$ can be arbitrary and the minimal loss at time $t$ may exceed $\alpha$ when $c_t > \alpha$. To prove boundedness of the sequence $\lambda_t$, we need to apply the regret bound on the AdaGrad-FTRL algorithm. The proof is presented in Appendix B.6.

**Theorem 2.6.** *Given any sequence of $R_1(\cdot), \ldots, R_T(\cdot) : [0,1] \mapsto [0,1]$, let $\{c_t\}_{t=1}^{T}$ denote the inner-level update given by the AdaGrad-FTRL algorithm. The outer-level update in Algorithm 1 guarantees that*

$$\frac{1}{T}\sum_{t=1}^{T}\left[c_t + \frac{1}{1-\beta}(R_t(\lambda_t) - c_t)_+\right]$$
$$\le \alpha + \frac{\sqrt{T+1}}{T}C_1 + \frac{1}{T}C_2,$$

*where*

$$C_1 = \max\left\{1, \frac{\beta}{1-\beta}\right\}\left(\frac{3}{4} + \frac{1}{4(1-\beta)}\right),$$

*and*

$$C_2 = \frac{\lambda_1/\gamma + (1-\beta)^{-1} - \alpha}{1-\beta}.$$

Importantly, this guarantee holds *regardless of the quality of the threshold sequence* $\{c_t\}$. However, the choice of $\{c_t\}$ directly determines the *tightness* of the bound: if $c_t$ is far from the optimal threshold $c^*$, the surrogate objective may

be much larger than the true $\text{CVaR}_\beta$, leading to conservative behavior. Thus, while the outer CDT-level ensures validity, the role of the inner-level AdaGrad-FTRL update is to adaptively learn thresholds $c_t$ to avoid being overly conservative, as described in the previous section.

*Remark* 2.7. (Range of $\lambda_t$.) The update for $\lambda_t$ is intentionally unprojected. The interval $[\lambda_{\min}, \lambda_{\max}]$ is the natural range used for scaling and interpretation, while the theoretical analysis extends the loss outside this range. Theorem 2.6 shows that this extension does not lead to uncontrolled drift of the iterates.

## 3. Experiments

We evaluate the proposed RU Conformal CVaR control framework on two sequential decision-making problems: (i) toxicity control for LLM outputs and (ii) portfolio management under market distribution shift.

Across both experiments, we discard the first 100 rounds as a burn-in period. We report results on (i) realized $\widehat{\text{CVaR}}_\beta$ control, and (ii) the dynamics of the learned threshold $\lambda_t$.

### 3.1. Toxicity Control for LLM Outputs Experiment

**Environment.** We consider a pool of prompts $\{x_i\}$, each paired with multiple candidate LLM-generated responses $\{y_i^j\}_{j=1}^K$. Each response is annotated with: (i) a machine toxicity score $r_{m,i}^j \in [0,1]$ produced by a toxicity classifier, and (ii) a human toxicity score $r_i^j \in [0,1]$, which we treat as the ground-truth loss.

At each round $t$, one prompt is selected and $K$ candidate responses are sampled according to a time-varying sampling distribution (described below). The realized loss is the maximum human toxicity score among accepted responses after filtering out responses whose machine toxicity score exceeds $\lambda_t$. Note that this convention corresponds to the LLM abstaining from returning a response when all candidates are filtered out; if no response is shown, the toxicity loss is zero.

**Action parameter.** The controller selects a threshold $\lambda_t \in [0,1]$ and rejects all candidate responses whose machine toxicity score exceeds $\lambda_t$. This follows the same deployment protocol as in Chen et al. (2025).

**Models and datasets.** Following Chen et al. (2025), we create an inexpensive semi-synthetic benchmark using an existing machine scoring model as the "human annotator," and a biased scoring model as the "machine assessor." Specifically, we use the Detoxify model (Hanu & Unitary team, 2020) for $r(\cdot)$ and retrain the Detoxify model for $r_m(\cdot)$ on a biased subset of the Jigsaw Unintended Bias in Toxicity Classification dataset (cjadams et al., 2019) that consists of

the 15% most and least toxic instances.

We conduct experiments using Llama 3.2-3B (Meta AI, 2024). We draw prompts from the REALTOXICITYPROMPTS dataset (Gehman et al., 2020) using the sampling regimes described in the next section.

**Synthetic distribution shift via tail thickening.** To simulate controlled non-stationary and adversarial shifts in the tail behavior, we generate responses using a time-varying tail-severity parameter.

At each time $t$, we sample a quantile level

$$p_t \sim \text{Beta}(a_t, b_t),$$

which controls how extreme the selected response is within the candidate pool. Given $K$ candidate responses for the current prompt, we sort them by machine toxicity score and select the response with rank

$$k_t = \min\{K, \max\{1, \lceil K\, p_t \rceil\}\}.$$

Thus, small values of $p_t$ select benign responses, while values of $p_t$ near 1 select highly toxic responses.

We consider three Beta distribution sampling regimes: (i) *uniform* regime, with $a_t = b_t = 1$, (ii) *adversarial/non-stationary* regime with time-varying $a_t$ and fixed $b_t = 1$, and (iii) *adversarial- jumps* regime, where $b_t = 1$, and $a_t \in [0.5, 5]$ changes over time with random spikes and dips throughout the sampling window. Our adversarial constructions allow the sampling distribution to gradually concentrate more mass near 1 as time passes, thereby increasingly selecting high machine-toxicity responses over time and inducing a controlled tail-thickening distribution shift in regime (ii), and a volatile shift in regime (iii). Visualizations of the sampling distributions of both regimes can be found in Appendix C.

### 3.2. Results for Toxicity Control for LLM Outputs

We evaluate the proposed RU Conformal CVaR (RUCC) controller in the three sampling regimes: *uniform*, *adversarial*, and *adversarial-jump*. In all three settings, the target risk level is set to $\alpha = 0.1$ with $\gamma = 0.05$, and we report results for $\beta = 0.85$ in Figure 2. For results of $\beta \in \{0.75, 0.8, 0.85, 0.9\}$, see Appendix C.

**Evolution of the empirical $\text{CVaR}_\beta$.** The top row of Figure 2 displays the per-step realized empirical $\text{CVaR}_\beta$ trajectories of our RUCC controller (blue line), with the corresponding average realized $\text{CVaR}_\beta$ (orange dotted line) and the target risk level (red dashed line). Results are shown for the uniform regime (left panel), adversarial regime (middle panel), and adversarial-jump regime (right panel).

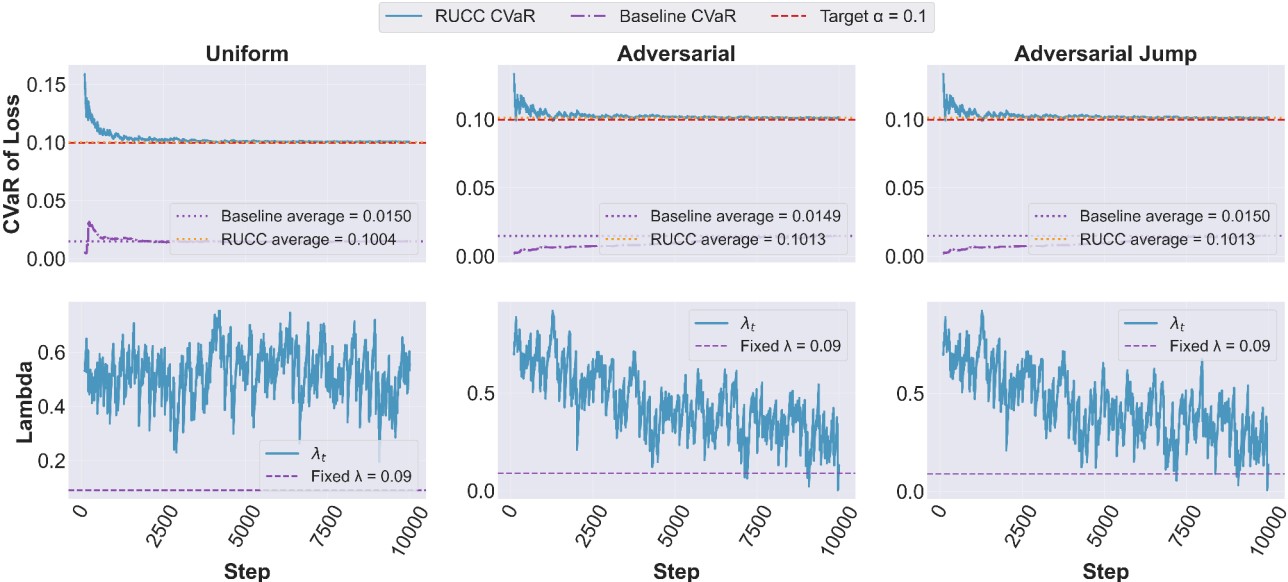

*Figure 2.* **LLM Toxicity Control Experiment.** Realized empirical $\mathrm{CVaR}_\beta$ and evolution of $\lambda_t$ for $\beta = 0.85$ with target $\alpha = 0.1$, step size $\gamma = 0.05$, and 100 steps of burn-in for the *Uniform* regime in the first column, *Adversarial* regime in middle column, and *Adversarial-Jump* regime in the right column.

We compare RUCC against a static baseline with a fixed $\lambda$ throughout the experiment. Specifically, we estimate a constant threshold $\hat{\lambda}$ using distortion risk control via L-statistics calibrated on the first 1000 samples in our sample set (Chen et al., 2025). For this baseline, we report the realized empirical $\mathrm{CVaR}_\beta$ trajectory (purple dashed line) and its average realized $\mathrm{CVaR}_\beta$ (purple dotted line).

From the top panel of Figure 2, we observe that the RUCC controller rapidly stabilizes the realized empirical $\mathrm{CVaR}_\beta$, keeping it close to the target level across all three settings. In the uniform regime, the trajectory settles relatively smoothly near the target. In contrast, the adversarial and adversarial-jump regimes exhibit more stochastic fluctuations, reflecting the greater difficulty of controlling tail risk when the data-generating process changes over time.

The figure also highlights the limitation of the baseline which selects a very small fixed value of $\hat{\lambda}$ that keeps $\mathrm{CVaR}_\beta$ well below the target, making the LLM severely overly conservative. By contrast, RUCC adjusts $\lambda_t$ over time to maintain tail risk control while avoiding excessive conservatism.

**Evolution of $\lambda_t$.** Next, we examine the evolution of $\lambda_t$ in the LLM toxicity control task under the uniformly sampled prompt stream and the adversarially sampled streams, shown in the bottom panel of Figure 2.

In the uniform regime, $\lambda_t$ remains relatively stable throughout the trajectory, exhibiting only mild fluctuations. This behavior is consistent with the underlying loss distribution

being comparatively stationary, so the algorithm does not need to adapt the threshold aggressively in order to maintain the target $\mathrm{CVaR}_\beta$ level.

In contrast, under the adversarial and adversarial-jump regimes, $\lambda_t$ decreases substantially over time and exhibits noticeably larger fluctuations than in the uniform setting. This reflects the increased difficulty of the control problem under distribution shift, where the underlying loss distribution changes over time. Nevertheless, despite these non-stationary environments, RUCC continues to adapt effectively and maintains robust control of the realized $\mathrm{CVaR}_\beta$.

### 3.3. Portfolio Management under Distribution Shift Experiment

**Environment.** We consider a two-asset portfolio consisting of (i) a risk-free asset (10-Year Treasury total-return index), and (ii) a risky asset (S&P 500 index).

**Action parameter.** Let $P_t$ denote the asset price at time $t$. We define the $h = 1$ log-return as follows,

$$r_t^{(1)} = \log \frac{P_t}{P_{t-1}}.$$

At each time $t$, the controller selects a portfolio weight $\lambda_t \in [0, 1]$, representing the fraction invested in the risky asset. The portfolio return is then,

$$r_t^p(\lambda_t) = \lambda_t r_t^{\text{risky}} + (1 - \lambda_t) r_t^{\text{risk-free}},$$

and the loss is defined as

$$R_t(\lambda_t) = -r_t^p(\lambda_t),$$

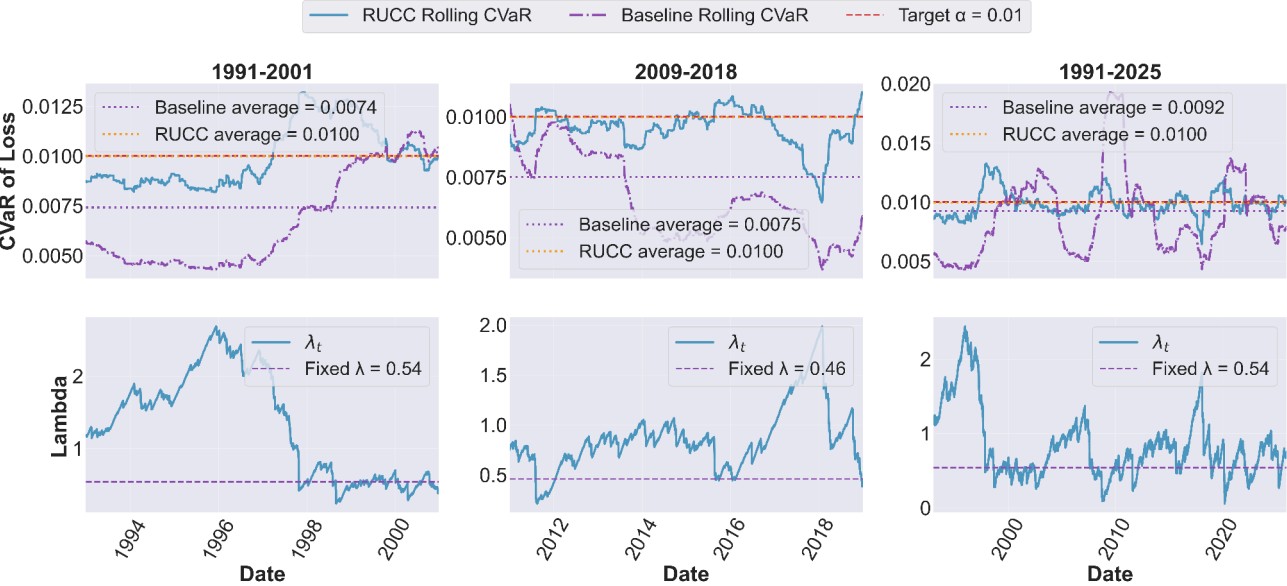

*Figure 3.* **Portfolio Management Experiment.** Realized empirical 180-day rolling $\mathrm{CVaR}_\beta$ (top row) and evolution of $\lambda_t$ (bottom row) for $\beta = 0.85$, $\gamma = 0.05$ with target $\alpha = 0.01$ and burn-in period of 100 days during 1991–2001 in the first column, 2009–2018 in second column, 1991–2025 in the third column.

where $r_t^{\mathrm{risky}}$ and $r_t^{\mathrm{risk\text{-}free}}$ denote the return of the risky asset and the return of the risk-free asset, respectively.

**Distribution shift.** We evaluate performance across market regimes from 1991 to 2025, covering both periods of economic stability and multiple market shocks including the dot-com crash, the 2008 financial crisis, COVID-19, and the subsequent high-inflation period.

**Datasets.** For our risky asset, we use historical S&P 500 index data via the `yfinance` API (Aroussi, 2020), which retrieves data from Yahoo Finance. For our risk-free asset, we use the 10-Year Treasury total-return index from the Federal Reserve Economic Data (FRED) database (Federal Reserve Bank of St. Louis, 2024).

### 3.4. Results for Portfolio Management Experiment

We evaluate the proposed RU Conformal CVaR controller on the portfolio management task across three time horizons: 1991–2001, 2009–2018, and 1991–2025. The 1991–2001 period corresponds to a relatively stable economic environment, while 2009–2018 captures a substantially more volatile regime that includes the global financial crisis and its aftermath. The full 1991–2025 horizon evaluates the long-term robustness of our proposed method across multiple market conditions.

Figure 3 reports the realized empirical 180-day rolling $\mathrm{CVaR}_\beta$ (top row) and the corresponding control parameter $\lambda_t$ (bottom row), for $\beta = 0.85$, target risk level $\alpha = 0.01$,

and step size $\gamma = 0.05$. The rolling-window CVaR is included as a diagnostic for the local behavior of the controller. Additional results for $\beta \in \{0.75, 0.80, 0.85, 0.90\}$ are provided in Appendix D.

**Evolution of the rolling empirical** $\mathrm{CVaR}_\beta$. The top row of Figure 3 displays the 180-day rolling realized empirical $\mathrm{CVaR}_\beta$ trajectories produced by the RUCC controller (blue line), together with the corresponding average realized $\mathrm{CVaR}_\beta$ (orange dotted line) and the target risk level (red dashed line) for $\beta = 0.85$.

We compare RUCC against a static baseline in which the control parameter $\lambda$ is fixed throughout the experiment. Specifically, we tune $\lambda$ over a grid of candidate values and select a fixed $\hat{\lambda}$ that controls the realized empirical $\mathrm{CVaR}_\beta$ over the first 1000 days of the sample period. For this baseline, we report both the 180-day rolling realized empirical $\mathrm{CVaR}_\beta$ trajectory (purple dashed line) and its average realized $\mathrm{CVaR}_\beta$ (purple dotted line).

The three evaluation windows: 1991–2001, 2009–2018, and 1991–2025, are shown in the left, middle, and right panels of the top row of Figure 3, respectively. Across all three periods, the RUCC controller consistently maintains the realized empirical $\mathrm{CVaR}_\beta$ close to the target level, whereas the static baseline exhibits substantially different behavior across market regimes.

During both the relatively stable 1991–2001 period, and the more volatile 2009–2018 period, the baseline is overly conservative, producing realized $\mathrm{CVaR}_\beta$ values substan-

tially below the target level. These results illustrate the difficulty of selecting a single fixed value of $\lambda$ that performs well across changing market conditions. By contrast, the RUCC controller adaptively adjusts $\lambda_t$ over time and is able to maintain stable tail-risk control across both stable and volatile environments.

Moreover, over the full 1991–2025 horizon, the method remains robust through multiple periods of economic stress. Although the realized $\mathrm{CVaR}_\beta$ increases modestly during episodes such as the dot-com bubble, the 2008 financial crisis, and the post-COVID market turbulence beginning in 2020, the rolling average empirical $\mathrm{CVaR}_\beta$ remains close to the target risk level throughout the sample period.

**Evolution of $\lambda_t$.** The bottom row of Figure 3 shows the evolution of $\lambda_t$, which represents the fraction of wealth allocated to the risky asset. During the 1991–2001 period, $\lambda_t$ exhibits a gradual long-term decline, indicating that the controller progressively reduces exposure to the risky asset over time while still maintaining moderate market participation.

In the 2009–2018 period, the controller initially maintains a relatively low value of $\lambda_t$ after the 2008 financial crisis and subsequently increases exposure as market conditions stabilize and the economy recovers. Over the full 1991–2025 horizon, the largest downward movements in $\lambda_t$ coincide with major episodes of market stress, including the 2008 financial crisis and the post-COVID turbulence beginning in 2020. This behavior illustrates that the controller dynamically responds to elevated tail-risk realizations by reducing risky exposure during periods of heightened uncertainty.

Note that the update for $\lambda_t$ is unconstrained, so $\lambda_t$ may temporarily leave the range $[0, 1]$. When evaluating portfolio decisions, we clip the realized allocation to the corresponding boundary value, so $\lambda_t < 0$ means zero risky-asset exposure and $\lambda_t > 1$ means maximal exposure. Thus, the realized portfolio allocation remains feasible. Theorem 2.6 shows that this unconstrained update cannot drift too far in aggregate, i.e., the cumulative excess over the target is only $O(\sqrt{T})$.

Overall, these experiments demonstrate that our method is capable of maintaining tail-risk control over multi-decade horizons spanning multiple crisis regimes while adaptively adjusting portfolio exposure over time. The results provide strong empirical evidence that the algorithm is robust to severe non-stationarity in the long run, achieving tight empirical $\mathrm{CVaR}_\beta$ control in realistic financial environments.

## 4. Discussion and Extensions

A key lever in our approach is the RU variational representation of $\mathrm{CVaR}_\beta$, which allows tail risk control to be reduced to online optimization over an auxiliary parameter. This idea is not unique to $\mathrm{CVaR}$. Many classical and modern risk measures admit similar variational representations, suggesting that our framework extends beyond tail risks. We describe illustrative examples below.

### 4.1. Optimized Certainty Equivalent (OCE) Risk Measures

A large class of risk measures known as *optimized certainty equivalents* (OCEs) (Ben-Tal & Teboulle, 2007) admit representations of the form

$$R(X) = \inf_{c \in \mathbb{R}} c + \mathbb{E}\big[\phi(X - c)\big]$$

for a suitable convex loss $\phi$. Recent work has shown how to control such risk measures using conformal methods in stationary, offline, and i.i.d. settings (Yeh et al., 2025). Our results suggest a clear extension to these guarantees in *online, non-stationary, and adversarial environments*.

### 4.2. Variational Risk Representations

The variance admits a well-known representation

$$\mathrm{Var}(X) = \mathbb{E}\big[(X - \mathbb{E}[X])^2\big] = \min_{a \in \mathbb{R}} \mathbb{E}\big[(X - a)^2\big],$$

which is widely used in quality control and robust estimation. This shows that variance control can likewise be interpreted as controlling the expectation of a surrogate loss indexed by an auxiliary parameter, making it amenable to our optimization framework.

## Impact Statement

This work contributes to the development of machine learning frameworks with increased safety and reliability by providing a distribution-free method for controlling tail risk in non-stationary and adversarial environments. By enabling online control of the Conditional Value-at-Risk ($\mathrm{CVaR}$), the proposed framework helps reduce the likelihood of rare but catastrophic failures in safety-critical applications such as financial decision-making and large language model deployment.

Beyond $\mathrm{CVaR}$, the framework provides a general template for controlling a broad class of nonlinear risk measures that admit a variational representation. We expect this contribution to help facilitate the design of more robust and trustworthy learning systems in environments subject to distribution shifts and strategic manipulation. We do not foresee negative societal impacts arising from this work; instead, its primary effect is to improve the safety and reliability of deployed machine learning systems.

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

*Table 1.* Summary of notation used throughout the paper.

| Symbol | Meaning |
| --- | --- |
| $t = 1, \ldots, T$ | Time index and total horizon |
| $R_t(\cdot)$ | Loss function revealed at round $t$ |
| $R_t = R_t(\lambda_t)$ | Realized loss at round $t$ |
| $R_{1:T}$ | Sequence of realized losses $(R_1, \ldots, R_T)$ |
| $\beta \in (0, 1)$ | CVaR level confidence level |
| $\alpha$ | Target CVaR risk level |
| $\lambda_t$ | Outer-update decision parameter (controls system action) |
| $\gamma$ | Step size for outer-update (CDT-style) |
| $c_t$ | Inner-update threshold parameter at round $t$ |
| $c^*$ | Optimal threshold in hindsight (RU minimizer) |
| $f_t(c)$ | RU surrogate loss: $c + \frac{1}{1-\beta}(R_t - c)_+$ |
| $F_T(c)$ | Empirical RU objective: $\frac{1}{T}\sum_{t=1}^{T} f_t(c)$ |
| $\widehat{\mathrm{CVaR}}_\beta(R_{1:T})$ | Empirical CVaR of realized sequence |
| $\hat{q}_\beta(R)$ | Empirical $\beta$-quantile of $\{R_1, \ldots, R_T\}$ |
| $\widehat{\mathbb{E}}_T[\cdot]$ | Empirical average over $t = 1, \ldots, T$ |
| $(x)_+$ | Positive part: $\max\{x, 0\}$ |
| $\mathrm{Reg}_T(c)$ | Cumulative RU regret of inner update |
| $g_t$ | Subgradient of $f_t$ at $c_t$ |
| $\mathbf{1}\{\cdot\}$ | Indicator function |

# A. Appendix

## A.1. Notation

For convenience, we summarize all notation used throughout the paper in Table 1. All quantities with a hat denote empirical (finite-sample) versions. Unless stated otherwise, all sequences $\{R_t\}_{t=1}^{T}$ may be non-stationary, or adversarial.

# B. Proofs

As discussed at the beginning of Section 2.4, we can transform $\lambda$ and $R(\cdot)$ by (5) and (6), and assume

$$\lambda_{\min} = 0, \quad \lambda_{\max} = 1, \quad R_{\min} = 0, \quad R_{\max} = 1, \quad \alpha \in (0, 1).$$

## B.1. Proof of Proposition 2.1

Define the RU objective

$$f_t(c) \;=\; c + \frac{1}{1-\beta}(R_t - c)_+, \qquad F_T(c) \;=\; \frac{1}{T}\sum_{t=1}^{T} f_t(c).$$

By the RU variational representation (applied to the realized sequence $R_{1:T}$),

$$\widehat{\mathrm{CVaR}}_\beta(R_{1:T}) \;=\; \min_{c \in \mathbb{R}} F_T(c).$$

Since $R_t \in [0, 1]$, $f_t$ is decreasing on $(-\infty, 0]$ and increasing on $[1, \infty)$. Thus,

$$\widehat{\mathrm{CVaR}}_\beta(R_{1:T}) \;=\; \min_{c \in [0,1]} F_T(c). \tag{8}$$

By the regret definition,

$$\frac{1}{T}\sum_{t=1}^{T}f_t(c_t) = \min_{c\in[0,1]}F_T(c) + \frac{\max_{c\in[0,1]}\text{Reg}_T(c)}{T} \tag{9}$$

$$= \widehat{\text{CVaR}}_\beta(R_{1:T}) + \frac{\max_{c\in[0,1]}\text{Reg}_T(c)}{T}. \tag{10}$$

Under (4),

$$\frac{1}{T}\sum_{t=1}^{T}f_t(c_t) \leq \alpha + o(1). \tag{11}$$

Combining (9) and (11) yields

$$\widehat{\text{CVaR}}_\beta(R_{1:T}) \leq \alpha + o(1) - \frac{\max_{c\in[0,1]}\text{Reg}_T(c)}{T}.$$

By assumption (3), uniformly over $c \in [0,1]$,

$$\frac{|\max_{c\in[0,1]}\text{Reg}_T(c)|}{T} = o(1).$$

Hence

$$\widehat{\text{CVaR}}_\beta(R_{1:T}) \leq \alpha + o(1),$$

which proves (1).

**B.2. Proof of Theorem 2.2**

By (9),

$$\widehat{\text{CVaR}}_\beta(R_{1:T}) \leq \alpha + \left(\frac{1}{T}\sum_{t=1}^{T}f_t(c_t) - \alpha\right) + \frac{|\max_{c\in[0,1]}\text{Reg}_T(c)|}{T}.$$

The proof is then completed by applying Theorems 2.4 and 2.6.

**B.3. Proof of Theorem 2.4**

We prove a slightly more general version that only requires $q_0 \geq \max\{1, \beta/(1-\beta)\}^2$. Let

$$F_t(c) = \sum_{s=1}^{t}f_s(c), \qquad \psi_t(c) = \frac{1}{2\eta_t}\left(c - \frac{1}{2}\right)^2, \qquad M_t = \min_{c\in[0,1]}\{F_t(c) + \psi_t(c)\}.$$

Then $c_{t+1}$ minimizes $F_t + \psi_t$, and $M_0 = 0$.

We first prove the lower bound. Since $c_t$ minimizes $F_{t-1} + \psi_{t-1}$,

$$M_t \leq F_t(c_t) + \psi_t(c_t)$$
$$= M_{t-1} + f_t(c_t) + \psi_t(c_t) - \psi_{t-1}(c_t).$$

The sequence $(\eta_t)$ is nonincreasing and $|c_t - 1/2| \leq 1/2$, so

$$\psi_t(c_t) - \psi_{t-1}(c_t) \leq \frac{1}{8}\left(\frac{1}{\eta_t} - \frac{1}{\eta_{t-1}}\right).$$

Thus

$$f_t(c_t) \geq M_t - M_{t-1} - \frac{1}{8}\left(\frac{1}{\eta_t} - \frac{1}{\eta_{t-1}}\right).$$

Summing over $t$ and using $M_T \geq \min_{c \in [0,1]} F_T(c)$ gives

$$\sum_{t=1}^{T} f_t(c_t) \geq \min_{c \in [0,1]} F_T(c) - \frac{1}{8}\left(\frac{1}{\eta_T} - \frac{1}{\eta_0}\right) = \min_{c \in [0,1]} \sum_{t=1}^{T} f_t(c) - \frac{\sqrt{q_T} - \sqrt{q_0}}{4}.$$

We now prove the upper bound. The function $F_{t-1} + \psi_{t-1}$ is $1/\eta_{t-1}$-strongly convex and is minimized at $c_t$. Hence, for every $c \in [0,1]$,

$$F_{t-1}(c) + \psi_{t-1}(c) \geq M_{t-1} + \frac{1}{2\eta_{t-1}}(c - c_t)^2.$$

By convexity of $f_t$,

$$f_t(c) \geq f_t(c_t) + g_t(c - c_t).$$

Since $\psi_t \geq \psi_{t-1}$, we obtain

$$M_t = \min_{c \in [0,1]} \{F_{t-1}(c) + \psi_{t-1}(c) + f_t(c) + \psi_t(c) - \psi_{t-1}(c)\}$$

$$\geq M_{t-1} + f_t(c_t) + \min_{c \in [0,1]} \left\{\frac{1}{2\eta_{t-1}}(c - c_t)^2 + g_t(c - c_t)\right\}$$

$$\geq M_{t-1} + f_t(c_t) - \frac{\eta_{t-1}}{2}g_t^2.$$

Therefore

$$f_t(c_t) \leq M_t - M_{t-1} + \frac{\eta_{t-1}}{2}g_t^2.$$

After summing,

$$\sum_{t=1}^{T} f_t(c_t) \leq M_T + \frac{1}{2}\sum_{t=1}^{T}\eta_{t-1}g_t^2.$$

Let $u \in \arg\min_{c \in [0,1]} F_T(c)$. Since $|u - 1/2| \leq 1/2$,

$$M_T \leq F_T(u) + \psi_T(u) \leq \min_{c \in [0,1]} F_T(c) + \frac{1}{8\eta_T} = \min_{c \in [0,1]} \sum_{t=1}^{T} f_t(c) + \frac{\sqrt{q_T}}{4}.$$

It remains to bound the AdaGrad sum more sharply. For $x = q_{t-1}$ and $a = g_t^2$,

$$\frac{a}{\sqrt{x}} = 2(\sqrt{x+a} - \sqrt{x}) + \sqrt{x}\left(\sqrt{1 + \frac{a}{x}} - 1\right)^2.$$

Let $s = \sqrt{1 + a/x}$. Since $a \leq q_0 \leq x$,

$$xs(s - 1) = x + a - \sqrt{x(x+a)} \leq a \leq q_0.$$

Rearranging it implies

$$\sqrt{x}\left(\sqrt{1 + \frac{a}{x}} - 1\right)^2 \leq q_0\left(\frac{1}{\sqrt{x}} - \frac{1}{\sqrt{x+a}}\right).$$

Consequently,

$$\frac{g_t^2}{\sqrt{q_{t-1}}} \leq 2(\sqrt{q_t} - \sqrt{q_{t-1}}) + q_0\left(\frac{1}{\sqrt{q_{t-1}}} - \frac{1}{\sqrt{q_t}}\right).$$

Summing over $t$ gives

$$\sum_{t=1}^{T} \frac{g_t^2}{\sqrt{q_{t-1}}} \leq 2(\sqrt{q_T} - \sqrt{q_0}) + q_0\left(\frac{1}{\sqrt{q_0}} - \frac{1}{\sqrt{q_T}}\right)$$

$$= 2\sqrt{q_T} - \sqrt{q_0} - \frac{q_0}{\sqrt{q_T}}.$$

Since $\eta_{t-1} = 1/(2\sqrt{q_{t-1}})$,

$$\frac{1}{2}\sum_{t=1}^{T}\eta_{t-1}g_t^2 = \frac{1}{4}\sum_{t=1}^{T}\frac{g_t^2}{\sqrt{q_{t-1}}}$$

$$\leq \frac{1}{2}\sqrt{q_T} - \frac{1}{4}\sqrt{q_0} - \frac{q_0}{4\sqrt{q_T}}.$$

Combining the preceding displays yields

$$\sum_{t=1}^{T}f_t(c_t) - \min_{c\in[0,1]}\sum_{t=1}^{T}f_t(c) \leq \frac{3}{4}\sqrt{q_T} - \frac{1}{4}\sqrt{q_0} - \frac{q_0}{4\sqrt{q_T}} \leq \frac{3}{4}\sqrt{q_T}, \tag{12}$$

which is the claimed upper bound.

Let

$$N_+(T) = \#\{t : g_t = 1\}, \qquad N_-(T) = \#\left\{t : g_t = -\frac{\beta}{1-\beta}\right\},$$

then

$$q_T = q_0 + N_+(T) + \left(\frac{\beta}{1-\beta}\right)^2 N_-(T)$$

$$\leq \max\left\{1, \left(\frac{\beta}{1-\beta}\right)^2\right\}(1 + N_+(T) + N_-(T))$$

$$= \max\left\{1, \left(\frac{\beta}{1-\beta}\right)^2\right\}(T+1).$$

$\square$

## B.4. Proof of Theorem 2.5

*Proof.* Let $R_t$ be independent with

$$\mathbb{P}(R_t = 0) = \beta, \qquad \mathbb{P}(R_t = 1) = 1 - \beta.$$

For $c \in [0,1]$ and $b = \frac{\beta}{1-\beta}$,

$$f_t(c) = c \quad \text{if } R_t = 0, \qquad f_t(c) = \frac{1}{1-\beta} - bc \quad \text{if } R_t = 1.$$

Thus the additive constants cancel in comparison with a fixed threshold. Define

$$g_t := \begin{cases} 1, & R_t = 0, \\ -b, & R_t = 1. \end{cases}$$

Then

$$f_t(c) = \text{constant} + g_t c,$$

and

$$\max_{c\in[0,1]}\text{Reg}_T(c) = \sum_{t=1}^{T}g_t c_t + \left(-\sum_{t=1}^{T}g_t\right)_+.$$

where the independent slopes satisfy

$$\mathbb{P}(g_t = 1) = \beta, \qquad \mathbb{P}(g_t = -b) = 1 - \beta.$$

They have mean zero. Since $c_t$ is chosen before $R_t$ is drawn,

$$\mathbb{E}[g_t c_t] = 0.$$

Therefore

$$\mathbb{E} \max_{c \in [0,1]} \text{Reg}_T(c) = \mathbb{E}\left[\left(-\sum_{t=1}^{T} g_t\right)_+\right].$$

If $N_T = \#\{t : g_t = 1\}$, then $N_T \sim \text{Binomial}(T, \beta)$ and

$$\sum_{t=1}^{T} g_t = N_T - b(T - N_T) = \frac{N_T - \beta T}{1 - \beta}.$$

Hence

$$\mathbb{E} \max_{c \in [0,1]} [\text{Reg}_T(c)] = \mathbb{E}\left[\left(-\frac{N_T - \beta T}{1 - \beta}\right)_+\right].$$

Thus, there exists a realization of $(R_t)_{t=1}^{T}$ such that

$$\max_{c \in [0,1]} \text{Reg}_T(c) \geq \mathbb{E}\left[\left(-\frac{N_T - \beta T}{1 - \beta}\right)_+\right].$$

Finally,

$$\frac{N_T - \beta T}{\sqrt{T\beta(1 - \beta)}} \Rightarrow Z, \qquad Z \sim N(0, 1),$$

and the normalized variables are uniformly integrable because their second moments are bounded. Thus

$$\frac{1}{\sqrt{T}}\mathbb{E}\left[\left(-\frac{N_T - \beta T}{1 - \beta}\right)_+\right] \to \frac{\sqrt{\beta(1 - \beta)}}{1 - \beta}\mathbb{E}[(-Z)_+] = \frac{1}{\sqrt{2\pi}}\sqrt{\frac{\beta}{1 - \beta}}.$$

$\square$

## B.5. Matching the lower bound with knowledge of $T$

**Theorem B.1.** *For every $T \geq 1$, there is a deterministic online learning algorithm with the knowledge of $T$, such that*

$$\max_{c \in [0,1]} \text{Reg}_T(c) \leq \frac{1}{\sqrt{2\pi}}\sqrt{\frac{\beta}{1 - \beta}}\sqrt{T} \cdot (1 + o(1)),$$

*for every sequence $R_1, \ldots, R_T \in [0, 1]$.*

*Proof.* Let $g_1, g_2, \ldots$ be independent with

$$\mathbb{P}(g_j = 1) = \beta, \qquad \mathbb{P}(g_j = -b) = 1 - \beta,$$

and define, for $n \geq 0$,

$$\Phi_n(s) = \mathbb{E}\left[\left(-s - \sum_{j=1}^{n} g_j\right)_+\right].$$

Set $S_0 = 0$. At round $t$, play

$$c_t = \frac{\Phi_{T-t}(S_{t-1} - b) - \Phi_{T-t}(S_{t-1} + 1)}{1 + b}.$$

After observing $R_t$, set

$$g_t = 1 - \frac{1}{1 - \beta}\mathbf{1}\{R_t > c_t\} \in \{1, -b\},$$

The choice $g_t = 1$ at $R_t = c_t$ is a valid subgradient because $\partial f_t(c_t) = [-b, 1]$ at the kink.

The function $\Phi_n$ is nonincreasing and 1-Lipschitz. Since $S_{t-1} - b \le S_{t-1} + 1$, the numerator defining $c_t$ is nonnegative and at most $1 + b$. Hence $c_t \in [0, 1]$. Also, since $\beta = b/(1 + b)$,

$$\Phi_{n+1}(s) = \frac{b\Phi_n(s+1) + \Phi_n(s - b)}{1 + b}.$$

The definition of $c_t$ gives, for either $g \in \{1, -b\}$,

$$gc_t + \Phi_{T-t}(S_{t-1} + g) = \Phi_{T-t+1}(S_{t-1}).$$

With $g = g_t$, this telescopes to

$$\sum_{t=1}^{T} g_t c_t + (-S_T)_+ = \Phi_T(0).$$

For every fixed $c \in [0, 1]$, convexity gives

$$f_t(c_t) - f_t(c) \le g_t(c_t - c).$$

Taking the maximum over $c \in [0, 1]$,

$$\max_{c \in [0,1]} \mathrm{Reg}_T(c) \le \sum_{t=1}^{T} g_t c_t - \min_{c \in [0,1]} c \sum_{t=1}^{T} g_t$$

$$= \sum_{t=1}^{T} g_t c_t + (-S_T)_+$$

$$= \Phi_T(0) = \mathbb{E}\left[\left(-\sum_{t=1}^{T} g_t\right)_+\right].$$

The upper bound follows from the same calculation in the proof of Theorem 2.5. $\square$

## B.6. Proof of Theorem 2.6

Recall that $R_t$ is extended outside of $[0, 1]$ with

$$R_t(\lambda) = \begin{cases} 0, & \lambda < 0, \\ 1, & \lambda > 1, \end{cases}$$

and the definition of $f_t$ in (7):

$$f_t(c) = c + \frac{1}{1 - \beta}\left(R_t(\lambda_t) - c\right)_+.$$

We start by proving a lemma.

**Lemma B.2.** *Let* $x_1, \ldots, x_s \in [0, 1]$ *and* $x_\tau = 0$ *for* $\tau \ge s + 1$. *Then, for every* $t \ge s$,

$$\min_{c \in [0,1]} \sum_{i=1}^{t} \left[c + \frac{(x_i - c)_+}{1 - \beta}\right] - \alpha t \le \frac{1}{1 - \beta}\left(\min_{c \in [0,1]} \sum_{i=1}^{s} \left[c + \frac{(x_i - c)_+}{1 - \beta}\right] - \alpha s\right)_+.$$

*Proof.* Let

$$V_t = \frac{1}{t} \min_{c \in [0,1]} \sum_{i=1}^{t} \left[c + \frac{(x_i - c)_+}{1 - \beta}\right].$$

For any $i \le s$ and $j > s$

$$\left[c + \frac{(x_i - c)_+}{1 - \beta}\right] \ge c = \left[c + \frac{(x_j - c)_+}{1 - \beta}\right].$$

Then we are left to prove

$$t(V_t - \alpha) \le \frac{s}{1-\beta}(V_s - \alpha)_+. \tag{13}$$

Thus, $V_t$ is nonincreasing in $t$ for $t \ge s$. In particular,

$$V_t \le V_s, \quad \forall t \ge s.$$

First suppose $(1-\beta)t \le s$. If $V_s \le \alpha$, then $V_t \le \alpha$ and (13) follows because the LHS is $t(V_t - \alpha) \le 0$ while the RHS is $s/(1-\beta)(V_s - \alpha)_+ = 0$. Otherwise, because $(1-\beta)t \le s$,

$$t(V_t - \alpha) \le \frac{s}{1-\beta}(V_s - \alpha) = \frac{s}{1-\beta}(V_s - \alpha)_+.$$

This proves the claim in the first case.

Now suppose $(1-\beta)t > s$. Let

$$c_t^* = \operatorname{argmin}_{c \in [0,1]} \sum_{i=1}^{t} \left[ c + \frac{(x_i - c)_+}{1-\beta} \right].$$

The first-order condition implies

$$0 \in t + \frac{1}{1-\beta} \sum_{i=1}^{t} \partial_c (x_i - c_t^*)_+.$$

If $c_t^* > 0$, then $\partial_c(x_i - c_t^*)_+ = 0$ for all $t > s$. Then

$$t + \frac{1}{1-\beta} \sum_{i=1}^{t} \partial_c(x_i - c_t^*)_+ = t + \frac{1}{1-\beta} \sum_{i=1}^{s} \partial_c(x_i - c_t^*)_+ \ge t - \frac{s}{1-\beta} > 0.$$

By contradiction, we conclude that $c_t^* = 0$. Therefore,

$$tV_t = \frac{1}{1-\beta} \sum_{i=1}^{t} x_i = \frac{1}{1-\beta} \sum_{i=1}^{s} x_i.$$

Thus,

$$t(V_t - \alpha) = \frac{1}{1-\beta} \sum_{i=1}^{s} x_i - \alpha t < \frac{1}{1-\beta} \left\{ \sum_{i=1}^{s} x_i - \alpha s \right\} \le \frac{1}{1-\beta} \left\{ \sum_{i=1}^{s} \left[ c + \frac{(x_i - c)_+}{1-\beta} \right] - \alpha s \right\} = \frac{s}{1-\beta}(V_s - \alpha).$$

Therefore the same bound follows in the second case. This proves the lemma. $\qquad\square$

**Proof of Theorem 2.6.** We now prove the theorem. If $\lambda_{T+1} \ge 0$, then the outer update gives

$$\sum_{t=1}^{T} f_t(c_t) - \alpha T = \frac{\lambda_1 - \lambda_{T+1}}{\gamma} \le \frac{\lambda_1}{\gamma}$$

which is already stronger than the claimed bound.

It remains to consider the case $\lambda_{T+1} < 0$. Let $s \le T$ be last time before $T+1$ at which $\lambda_s \ge 0$, i.e.,

$$\lambda_s \ge 0, \qquad \lambda_{s+1}, \lambda_{s+2}, \ldots, \lambda_{T+1} < 0.$$

Such an $s$ must exist as $\lambda_1 \ge 0$. At time $s$, $0 \le f_s(c_s) \le (1-\beta)^{-1}$. Therefore

$$\lambda_{s+1} = \lambda_s - \gamma(f_s(c_s) - \alpha) \ge -\gamma\big((1-\beta)^{-1} - \alpha\big).$$

Telescoping the outer update to time $s$ gives

$$\sum_{\tau=1}^{s} f_\tau(c_\tau) - \alpha s = \frac{\lambda_1 - \lambda_{s+1}}{\gamma} \le \frac{\lambda_1}{\gamma} + (1-\beta)^{-1} - \alpha.$$

Let

$$C_\beta = \frac{3}{4} \max\left\{1, \frac{\beta}{1-\beta}\right\}.$$

Using the lower bound for $\max_{c \in [0,1]} \mathrm{Reg}_T(c)$ in Theorem 2.4 to the first $s$ periods,

$$\min_{c \in [0,1]} \sum_{\tau=1}^{s} f_\tau(c) - \alpha s \leq \frac{\lambda_1}{\gamma} + (1-\beta)^{-1} - \alpha + \frac{C_\beta}{3}\sqrt{T+1}. \tag{14}$$

For every $t = s+1, \ldots, T$, we have $\lambda_t < 0$, hence $R_t(\lambda_t) = 0$. Thus the realized losses after time $s$ are zeros. By Lemma B.2,

$$\min_{c \in [0,1]} \sum_{\tau=1}^{T} f_\tau(c) - \alpha T \leq \frac{\lambda_1/\gamma + (1-\beta)^{-1} - \alpha + C_\beta \sqrt{T+1}}{1-\beta}.$$

Finally, using the upper bound on $\max_{c \in [0,1]} \mathrm{Reg}_T(c)$ in Theorem 2.4,

$$\sum_{t=1}^{T} f_t(c_t) - \alpha T \leq \min_{c \in [0,1]} \sum_{t=1}^{T} f_t(c) - \alpha T + C_\beta \sqrt{T+1}$$

$$\leq \frac{\lambda_1/\gamma + (1-\beta)^{-1} - \alpha}{1-\beta} + C_\beta \left(1 + \frac{1}{3(1-\beta)}\right)\sqrt{T+1}.$$

Dividing by $T$ proves the theorem.

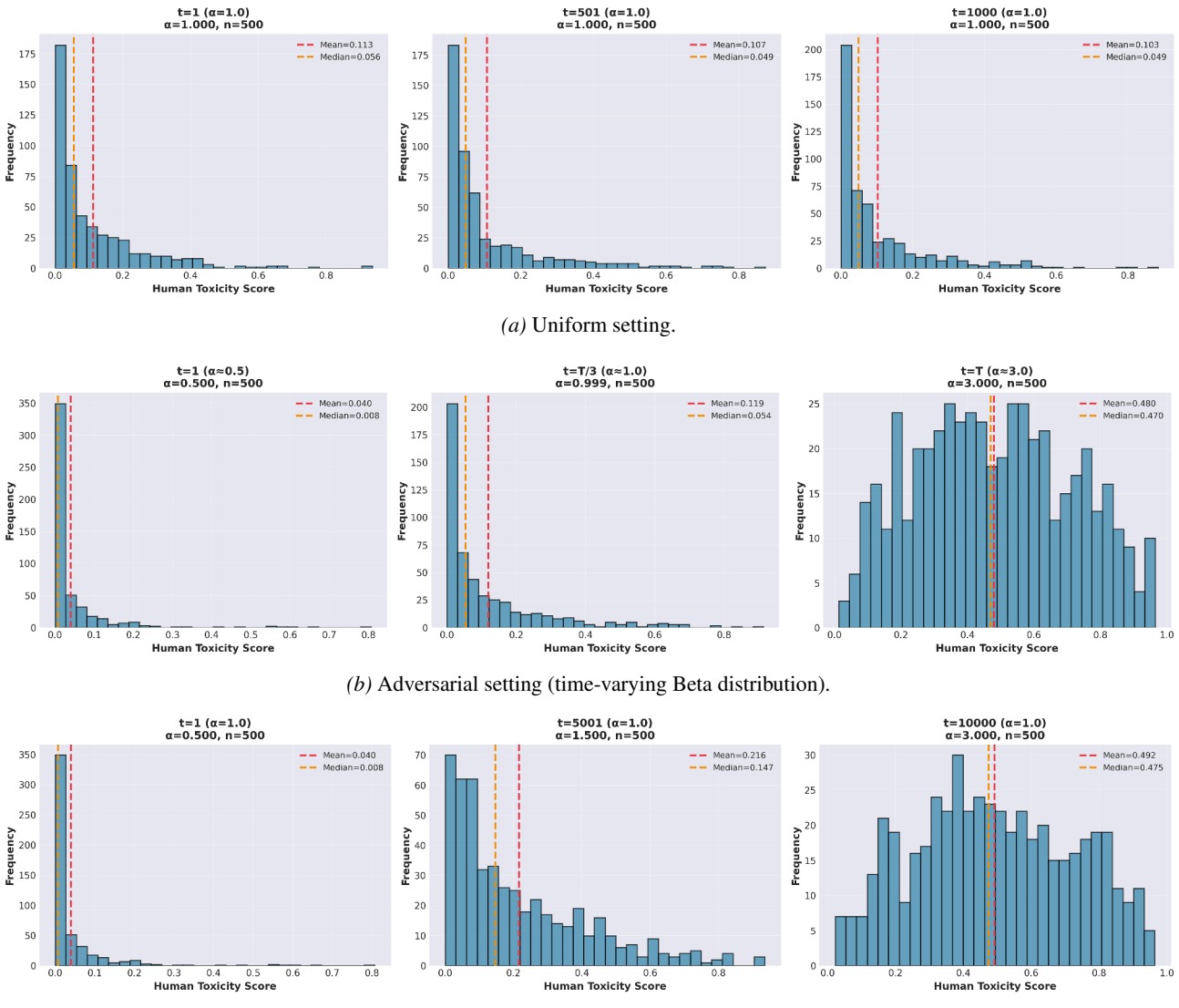

*(a)* Uniform setting.

*(b)* Adversarial setting (time-varying Beta distribution).

*(c)* Adversarial-jump setting (time-varying Beta distribution).

*Figure 4.* **LLM Toxicity Control Experiment.** Dataset toxicity distributions under different sampling regimes. (Top) Stationary uniform sampling. (Middle) Adversarial setting with progressively tail-thickening sampling. (Bottom) Adversarial-jump setting.

## C. Additional Results for Experiment 1: Large Language Model Toxicity Control

Figures 4a, 4b, and 4c visualize the empirical distribution of human toxicity scores for different values of the Beta sampling parameter $a$ under the uniform, adversarial, and adversarial-jump sampling regimes, respectively, and keeping $b$ fixed.

### C.1. Distributional Effects of the Sampling Parameter

**Uniform regime.** Figure 4a shows the corresponding distributions when $a_t = b_t = 1$. In this case, the toxicity distribution remains stable over time as the overall shape exhibits no systematic drift in either its center or tail behavior.

**Adversarial and adversarial-jump regime.** Figure 4b and 4c show the empirical toxicity distribution for several values of the Beta shape parameter $a_t$, with $b_t = 1$. As $a_t$ increases, the Beta distribution $c_t \sim \text{Beta}(a_t, b_t)$ concentrates more mass near 1, causing the sampling procedure to increasingly select more toxic responses. This results in a pronounced rightward shift and thickening of the upper tail of the toxicity distribution. Figure 5 illustrates the evolution of $a_t$ as a function of the time step for the adversarial and adversarial-jump sampling regime in the left and right panel, respectively.

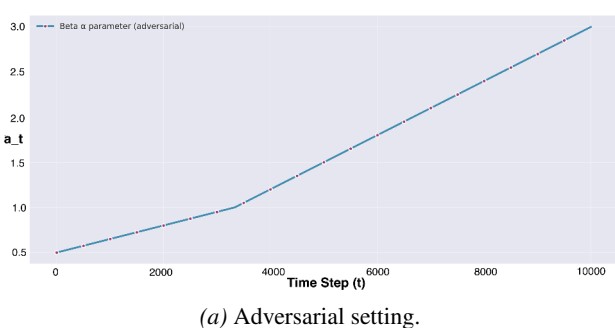
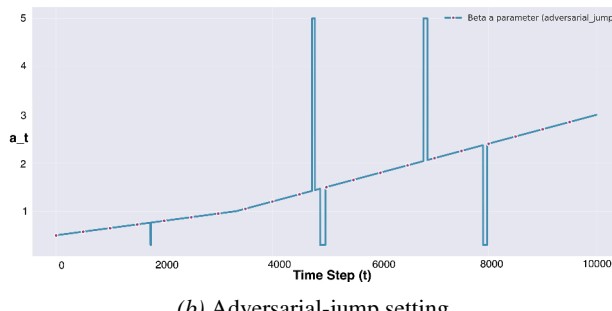

*(a)* Adversarial setting.    *(b)* Adversarial-jump setting.

*Figure 5.* **LLM Toxicity Control Experiment.** Evolution of $a_t$ parameter in Beta distribution under different sampling regimes: adversarial setting (left), adversarial-jump setting (right).

From the figures, it can be observed that $a_t$ is increasing overall across time for both regimes, while spontaneous shocks are injected to the adversarial-jump regime, where $a_t$ can take values as low as 0.5, or as high as 5.

## C.2. Additional Experiment Results

We evaluate the proposed RUCC controller under uniform, adversarial, and adversarial-jump sampling regimes, for tail levels $\beta \in \{0.75, 0.8, 0.85, 0.9\}$ with target risk level $\alpha = 0.1$. In all experiments, we use step size $\gamma = 0.05$ and discard the first 100 rounds as a burn-in period.

**Evolution of the empirical** $\text{CVaR}_\beta$**.**    The left panel of Figures 6, 7, and 8 show the evolution of the realized empirical $\text{CVaR}_\beta$ under the uniform, adversarial, and adversarial-jump regimes, respectively.

In the *uniform* setting (Figure 6), all curves exhibit a clear decay toward the target level $\alpha = 0.1$. Higher $\beta$ values (more risk-averse tail control) start from larger initial $\text{CVaR}_\beta$ values but converge at a comparable rate. After sufficient time, the realized $\text{CVaR}_\beta$ for all $\beta$ stabilizes tightly around the target level, demonstrating that the method can achieve tight, non-conservative control.

In the *adversarial* setting (Figure 7), and *adversarial-jump* setting (Figure 8), the controller adapts and drives the realized $\text{CVaR}_\beta$ downward for all $\beta$, eventually stabilizing near the target level. The convergence is slower and exhibits more fluctuation than in the uniform case, reflecting the intrinsic difficulty of controlling tail risk under adversarial distribution shift.

In all three settings, the RUCC controller is able to maintain tail risk control through adaptive $\lambda_t$ updates while avoiding overconservative behavior–which is demonstrated by the baseline realized $\text{CVaR}_\beta$ that selects a static $\lambda_t$.

*Remark* C.1. (Effect of the tail level $\beta$.) Across both regimes, larger $\beta$ values (corresponding to more extreme tail sensitivity) lead to systematically higher $\text{CVaR}_\beta$ trajectories, especially in the early stages. However, for all values of $\beta$, the empirical $\text{CVaR}_\beta$ eventually achieves stable control near $\alpha$.

**Evolution of the control parameter** $\lambda_t$**.**    The right panel of Figures 6, 7, and 8 show the evolution of the control parameter $\lambda_t$ in the uniform, adversarial, and adversarial-jump regimes, respectively.

In the *uniform* setting (Figure 6), $\lambda_t$ evolves within a comparatively constrained band, indicating a relatively stable environment. It can also be observed that the range of the band depends on $\beta$ in an increasing manner, demonstrating the inherent increase in difficulty of controlling risks at higher tails.

In the *adversarial* setting (Figure 7) and the *adversarial-jump* setting (Figure 8), $\lambda_t$ exhibits a clear decreasing trend over time for all $\beta$. Initially, $\lambda_t$ remains large, corresponding to less aggressive filtering. As the controller accumulates evidence and adapts to the realized tail distribution, $\lambda_t$ is gradually reduced, reflecting an increasingly strict control of the risky region. Larger $\beta$ values lead to systematically smaller $\lambda_t$, as expected for more stringent tail-risk control. Furthermore, in the *adversarial-jump* setting (Figure 8), in addition to the decreasing trend in $\lambda_t$, there are sporadic spikes in $\lambda_t$ that correspond to the dips in $a_t$ of the Beta sampling distribution, and the analogous dips in $\lambda_t$ that correspond to the jumps in $a_t$, demonstrating the robustness of the RUCC controller and its ability to adapt to sudden distributional changes, as well as general distribution drifts.

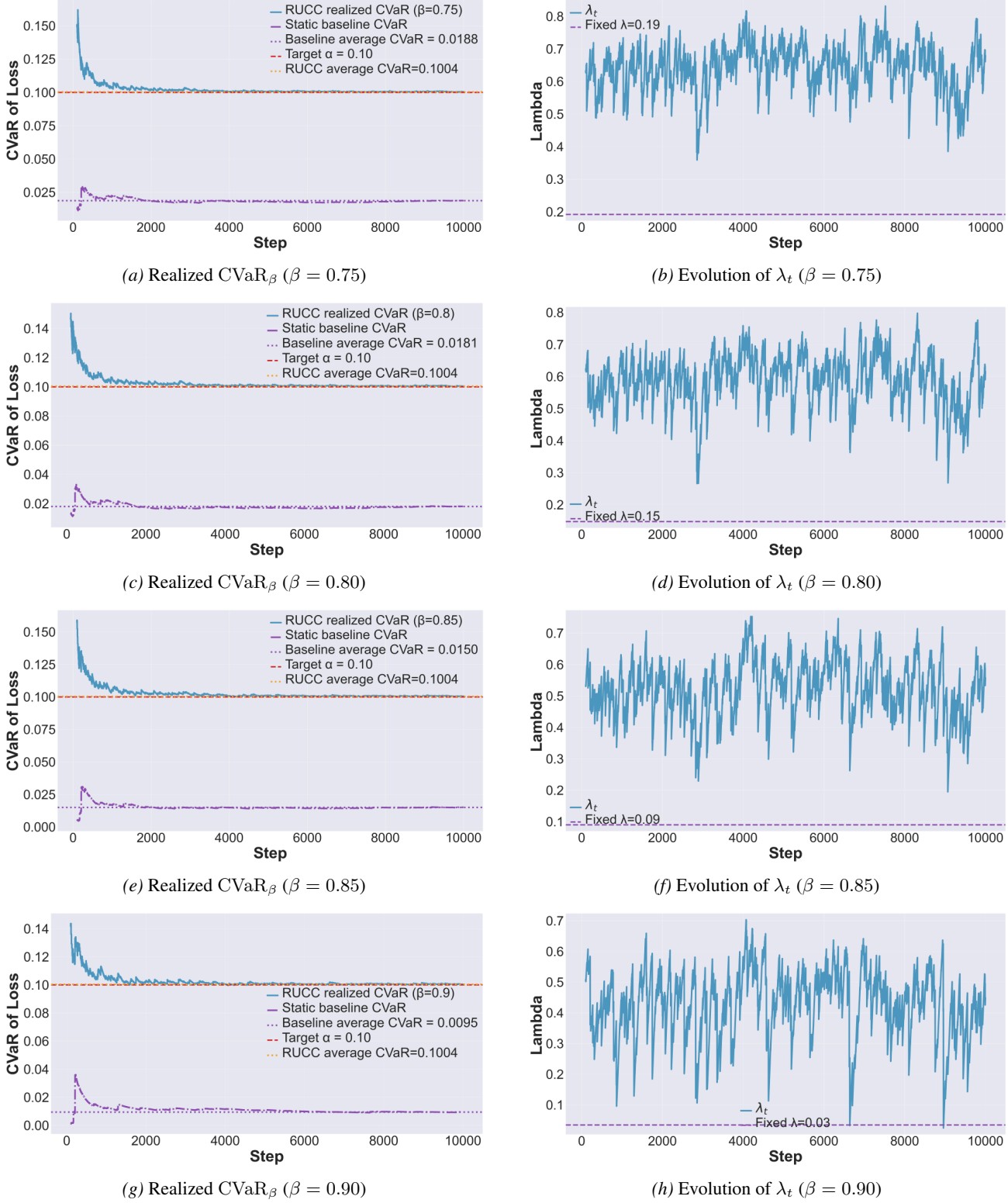

*Figure 6.* **LLM Toxicity Control Experiment.** Realized empirical $\mathrm{CVaR}_\beta$ and evolution of $\lambda_t$ with $\gamma = 0.05$, and target $\alpha = 0.1$ for the uniform regime

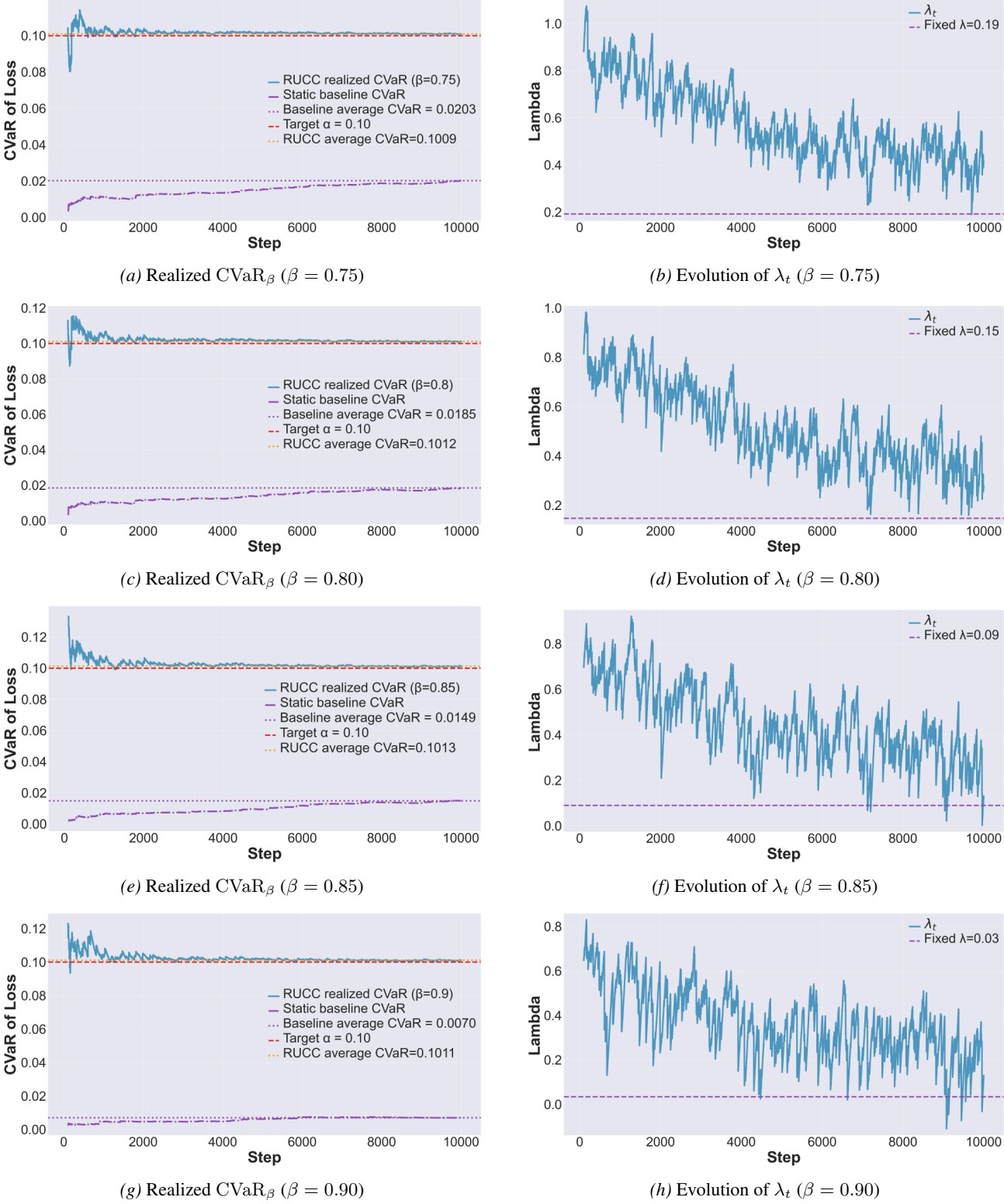

*(a)* Realized $\mathrm{CVaR}_\beta$ ($\beta = 0.75$)

*(b)* Evolution of $\lambda_t$ ($\beta = 0.75$)

*(c)* Realized $\mathrm{CVaR}_\beta$ ($\beta = 0.80$)

*(d)* Evolution of $\lambda_t$ ($\beta = 0.80$)

*(e)* Realized $\mathrm{CVaR}_\beta$ ($\beta = 0.85$)

*(f)* Evolution of $\lambda_t$ ($\beta = 0.85$)

*(g)* Realized $\mathrm{CVaR}_\beta$ ($\beta = 0.90$)

*(h)* Evolution of $\lambda_t$ ($\beta = 0.90$)

*Figure 7.* **LLM Toxicity Control Experiment.** Realized empirical $\mathrm{CVaR}_\beta$ and evolution of $\lambda_t$ with $\gamma = 0.05$, and target $\alpha = 0.1$ for the adversarial regime

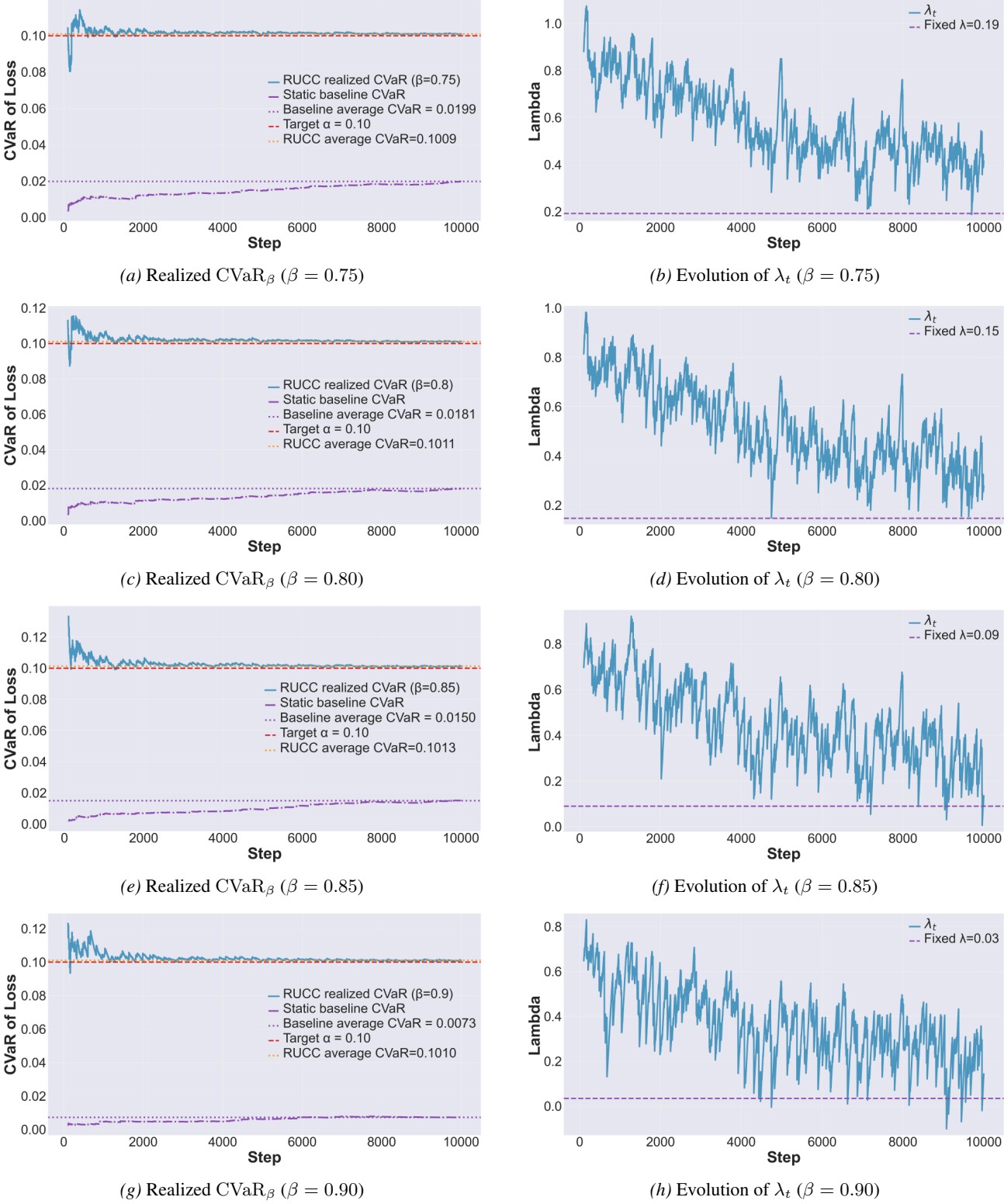

*Figure 8.* **LLM Toxicity Control Experiment.** Realized empirical $\mathrm{CVaR}_\beta$ and evolution of $\lambda_t$ with $\gamma = 0.05$, and target $\alpha = 0.1$ for the adversarial-jump regime

## D. Additional Results for Experiment 2: Portfolio Management

We evaluate the proposed RU Conformal CVaR controller on various horizons of real-market portfolio management tasks covering the period 1991–2001, 2009–2018, and the full period 1991–2026. We set the target risk level to $\alpha = 0.01$, and report results for $\beta \in \{0.75, 0.8, 0.85, 0.9\}$, with $\gamma = 0.05$, and remove the first 100 points as burn-in.

**Evolution of the rolling empirical** $\mathrm{CVaR}_\beta$. The left panel of Figures 9, 10, and 11 show the evolution of the 180-day rolling realized empirical $\mathrm{CVaR}_\beta$ for different $\beta$ during 1991–2001, 2009–2018, and the full period 1991–2025, respectively.

During 1991–2001, we observe that the $\mathrm{CVaR}_\beta$ of the baseline controller grows over time with a static $\lambda_t$ that eventually exceeds the target level. This phenomenon can be explained by the fact that 1991–2001 is an economically stable period with steady growth that ended with the dot-com crash in 2001. Accordingly, as the baseline controller has a fixed $\lambda_t$, its associated realized $\mathrm{CVaR}_\beta$ grows beyond the target level over time. In contrast, the realized tail risk of the RUCC controller initially starts below the target level and steadily increases over time as the controller adapts, and moves tightly around the target level as time passes.

During 2009–2018, we observe that the $\mathrm{CVaR}_\beta$ of the baseline controller decreases overtime with a static $\lambda_t$ that eventually falls well below the target level. This observation can be explained by the fact that 2009–2018 is the post-financial crisis period in which the economy is slowly improving over time (with decreasing risks). As the baseline controller has a fixed $\lambda_t$, its associated realized $\mathrm{CVaR}_\beta$ falls below the target level over time. In contrast, the realized tail risk of the RUCC controller initially starts below the target level (as a response to the financial crisis) and steadily increases over time as the controller adapts to the improving economic environment, to be close to the target tail risk level.

Finally, by observing the full-period 1991–2025, we can highlight the robust adaptability of the RUCC controller. While the realized tail risk of the baseline controller rises and falls drastically in response to the changing economic environment, including dramatic rises during the 2008 financial crisis and post-COVID period, the tail risk associated with the RUCC controller moves tightly around the target level with moderate fluctuations during the same economically tumultuous times.

In all cases, the final realized tail risk values are close to the target level (approximately 0.01), demonstrating that the method achieves tight long-run tail risk control without persistent conservatism. Higher $\beta$ values (corresponding to more extreme tail control) exhibit slightly slower convergence, as expected, but still reach the target level within the time horizon.

**Evolution of the control parameter** $\lambda_t$. The right panel of Figures 9, 10, and 11 shows the evolution of the control parameter $\lambda_t$ for different $\beta$.

During 1991–2001, we observe that the RUCC $\lambda_t$ grows then falls over time, indicating that the controller first increases then reduces exposure to the risky asset as tail risks accumulate. The initial increase in $\lambda_t$ is consistent with the economic growth during the period, the later decrease in $\lambda_t$ is consistent with the dot-com crash in 2000, that requires more stringent risk management to control the tail risk at the target level. On the other hand, the baseline fixed $\lambda$ stays at a comparatively low level causing it to be overly conservative.

In contrast, during 2009–2018, we observe that the RUCC $\lambda_t$ grows over time, indicating that the controller increases exposure to the risky asset as tail risks decrease. The growth in $\lambda_t$ over time is consistent with the growing stability of the economy after the 2008 financial crisis, allowing for more lenient risk management to control the tail risk at the desired target level. Conversely, the baseline fixed $\lambda$ stays at a consistently low level due to the financial crisis, and fails to adapt to the changing economic environment.

Finally, during 1991–2025, we observe that the RUCC $\lambda_t$ grows and falls over time, where $\lambda_t$ peaks during economically stable periods, such as pre-dot-com crash, pre-financial crisis, and pre-COVID. The larger values of $\lambda_t$ encourage investment in the risky asset while controlling the tail risk at the target level. On the other hand, $\lambda_t$ accordingly drops during economically tumultuous periods, such as post-dot-com crash, post-financial crisis, and post-COVID periods. The smaller values of $\lambda_t$ allow for less aggressive investment in the risky asset to avoid the burden of tail risks.

Note that in every time period for larger $\beta$, $\lambda_t$ shrinks accordingly as the algorithm must be more conservative in order to control more extreme tail events.

Overall, these results demonstrate that our proposed RU conformal inference controller successfully enforces long-run $\mathrm{CVaR}_\beta$ control on real financial data over a 25-year horizon, including multiple crisis periods (e.g., the dot-com crash, 2008 financial crisis, and COVID-19).

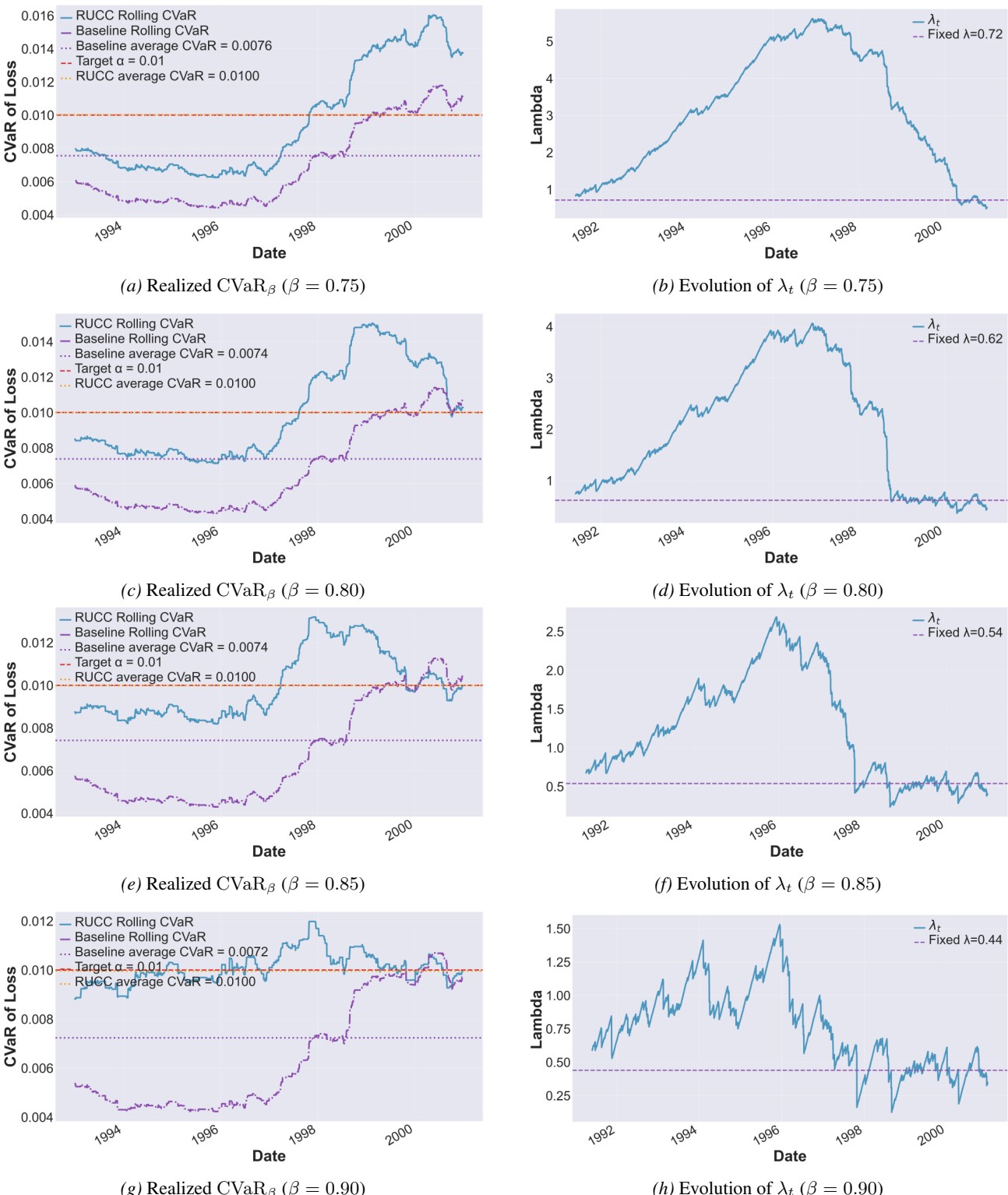

*Figure 9.* **Portfolio Management Experiment.** Realized empirical $\text{CVaR}_\beta$ and evolution of $\lambda_t$ for $\gamma = 0.05$ with target $\alpha = 0.01$ during 1991–2000.

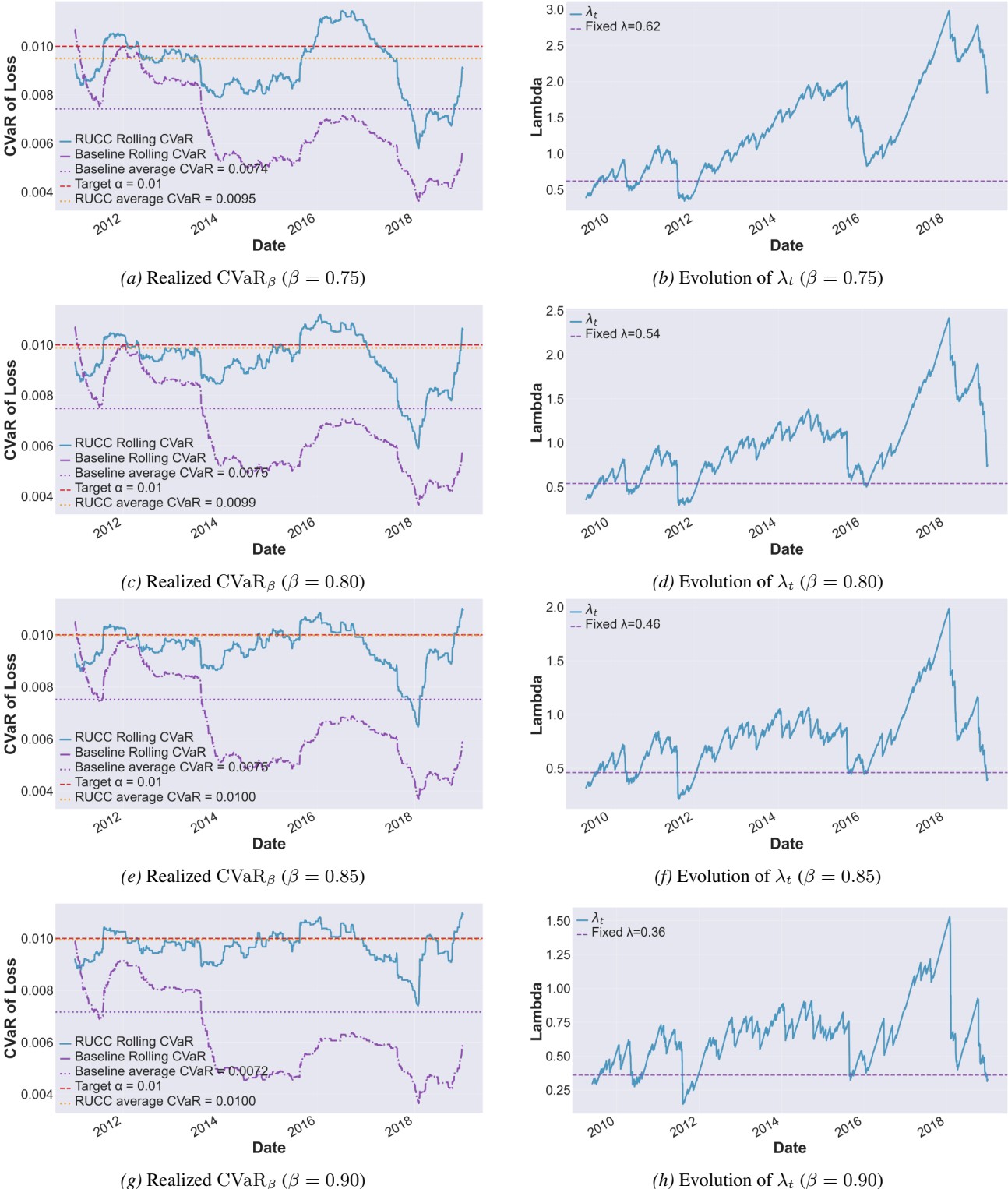

*(a)* Realized $\text{CVaR}_\beta$ ($\beta = 0.75$)

*(b)* Evolution of $\lambda_t$ ($\beta = 0.75$)

*(c)* Realized $\text{CVaR}_\beta$ ($\beta = 0.80$)

*(d)* Evolution of $\lambda_t$ ($\beta = 0.80$)

*(e)* Realized $\text{CVaR}_\beta$ ($\beta = 0.85$)

*(f)* Evolution of $\lambda_t$ ($\beta = 0.85$)

*(g)* Realized $\text{CVaR}_\beta$ ($\beta = 0.90$)

*(h)* Evolution of $\lambda_t$ ($\beta = 0.90$)

*Figure 10.* **Portfolio Management Experiment.** Realized empirical $\text{CVaR}_\beta$ and evolution of $\lambda_t$ for $\gamma = 0.05$ with target $\alpha = 0.01$ during 2009–2018.

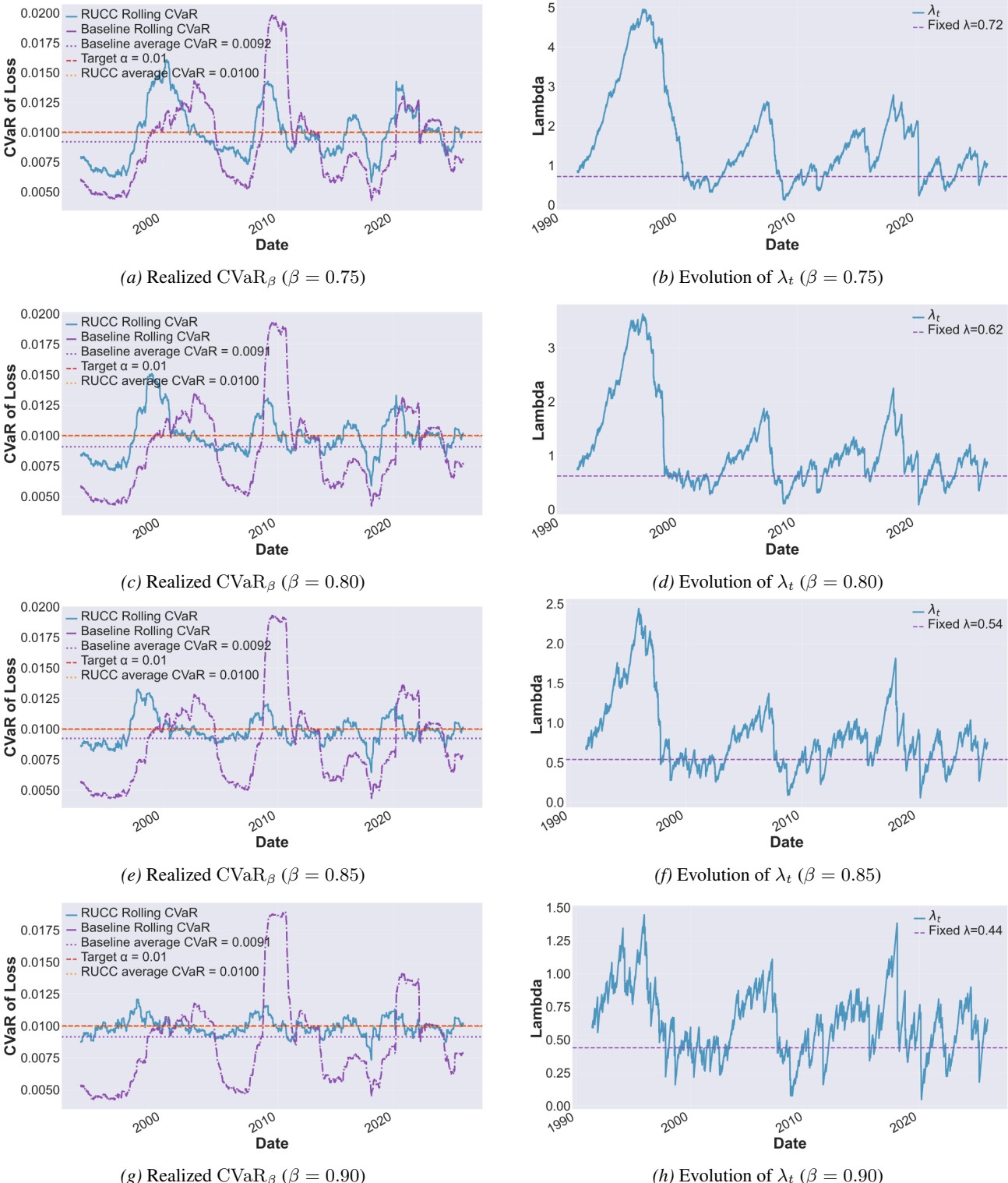

*Figure 11.* **Portfolio Management Experiment.** Realized empirical $\text{CVaR}_\beta$ and evolution of $\lambda_t$ for $\gamma = 0.05$ with target $\alpha = 0.01$ during 1991–2025.

