# OpenReview forum: "Adversarially Robust Control of Conditional Value-at-Risk via Rockafellar-Uryasev Conformal Inference"
_ICML.cc/2026/Conference — ICML 2026 regular_

### Official Review · Reviewer_Erzj · 2026-03-12

**Soundness:** 4
**Presentation:** 4
**Significance:** 3
**Originality:** 3
**Overall Recommendation:** 4
**Confidence:** 3

**Summary:**

This work proposes an elegant algorithm with theoretical guarantees for online controlling of the CVaR.

**Compliance With Llm Reviewing Policy:**

Affirmed.

**Key Questions For Authors:**

- What is eta in the statement of Proposition 2.4?
- How is the time-varying a_t determined in Section 3.1? Is alpha the same as a in Fig. 4?
- Can you comment on the dependence on B in Proposition 2.4?
- The notion of "nonstationarity" can be defined in different ways (e.g., several ways even in time series). Does it just mean time-dependent settings in this work?

**Limitations:**

see above

**Strengths And Weaknesses:**

strengths:
- The authors have proposed an elegant algorithm with strong guarantees.
- The presentation of the work is excellent, with great explanations of both the theory and experiments.
- The experiments are impressive, especially the one in Section 3.3, considering the multiple market shocks.

weaknesses:
- The main theorem 2.2 seems to rely on a sequence of existing results (Pro. 2.4 and 2.5). As a consequence, the theoretical contribution in this work might be a bit limited.

---

> ### Author Rebuttal · Authors · 2026-03-31
>
> We thank the reviewer for the careful reading and for the very positive assessment of the algorithm, theory, and experiments, especially the portfolio experiment under multiple market shocks.
>
> **Weakness:**
>
> We appreciate the concern regarding the reliance of Theorem 2.2 on Propositions 2.4 and 2.5. We agree that the current presentation may make the theoretical contribution appear more modular than intended, and we will revise the paper to clarify this point.
>
> Our main contribution is not the individual components, but the *reduction* from online CVaR control to a regret minimization problem via the Rockafellar–Uryasev representation, together with the nontrivial coupling of:
> - an outer CDT-style update controlling the RU surrogate, and
> - an inner no-regret update over the auxiliary threshold.
>
> This coupling yields an online CVaR control guarantee under adversarial/nonstationary data. We will make this novelty more explicit in the statement of contributions and around Theorem 2.2.
>
> In addition, we found that the unnormalized Kelly betting algorithm has a similar empirical performance and better theoretical property than the normalized Kelly betting algorithm we used in the current draft. Specifically, the inner-update is given by
> $$g_{t} \gets 1 - \frac{1}{1-\beta} \cdot \mathbf{1}\{ R_{t}(\lambda_{t}) > c_{t} \}.$$
> $$c_{t+1} \gets -\frac{1-\beta}{\beta}\frac{\sum_{i=1}^{t}g_i}{t+1} \left(\epsilon - \sum_{i=1}^{t}g_i c_i \right).$$
>
> For this new inner-loop update, we have proved a tighter version of Proposition 2.5 that pinned down the leading constant for the $O(\sqrt{\log T / T})$ regret rate. In particular, we show that the regret is $\sqrt{2(1-\beta)/\beta}\cdot \sqrt{\log T / T} (1 + o(1))$ when the loss function $R$ is bounded in $[0,1]$. This new proof leverages the special property of $ \nabla  f_t(c)$ that only takes two values.
>
> Furthermore, under a “bounded origin regret” assumption, introduced by Streeter and McMahan (2012), that restricts the candidate algorithm to have a bounded regret when the comparator takes c=0, we have proved a regret lower bound $(1-\epsilon) \sqrt{2(1-\beta)/\beta}\cdot \sqrt{\log T / T}$ for any $\epsilon > 0$ as $T\rightarrow \infty$.
>
> [1] Streeter, M., & McMahan, H. B. (2012). No-regret algorithms for unconstrained online convex optimization. arXiv preprint arXiv:1211.2260.
>
> ## Questions
>
> **Q1: What is $\eta$ in Proposition 2.4?**
> Thank you for catching this—this is a notation oversight in the current draft. We will correct and clarify the statement.
>
> **Q2: How is the time-varying $a_t$ determined? Is it the same as $\alpha$?**
> No. $\alpha$ is the target CVaR level, while $a_t$ is the time-varying shape parameter of the Beta distribution used to induce nonstationarity in the LLM experiment.
>
> In the adversarial regime, we vary $a_t$ over time (with $b_t = 1$) to place increasing mass near 1 and select more toxic responses. We will clarify both the distinction and the schedule in the revision (e.g., $a_t = 0.5$ at $t=1$, $a_t = 1.0$ at $t=T/3$, and $a_t = 3.0$ at $t=T$).
>
> **Q3: Dependence on $B$ in Proposition 2.4**
> The dependence on $B$ arises from the boundedness assumption on $R_t(\lambda)$, which controls the scale of the RU surrogate and stabilizes the outer update. The convergence rate in $T$ is unchanged; $B$ only affects constant factors. We will make this dependence explicit rather than hiding it in big-$O$ notation.
>
> **Q4: Meaning of “nonstationarity”**
> In this work, ‘nonstationary’ is used in the broad sequential sense: the loss functions / data-generating environment may vary over time, and may even depend adaptively on the past and on the learner’s current action. So yes, it includes time-dependent settings, but is strictly broader than a fixed parametric notion of drift from time-series analysis. We will clarify this terminology early in the paper to avoid ambiguity.

---

> > ### Author Rebuttal · Reviewer_Erzj · 2026-04-05
> >
> > I thank the authors for the detailed comments.

---

### Official Review · Reviewer_Qgpv · 2026-03-12

**Soundness:** 3
**Presentation:** 3
**Significance:** 3
**Originality:** 3
**Overall Recommendation:** 5
**Confidence:** 3

**Summary:**

The paper provides a method for controlling the empirical conditional variance of risk (CVaR) of losses $R_t(\lambda_t($ from a decision variable $\lambda_t$, when the sequence of losses is nonstationary.

It combines several techniques and results, most crucially the RU-representation of CVaR via a convex minimization problem.

The method is illustrated in two distinct applications, LLM toxicity control and portfolio management.

**Compliance With Llm Reviewing Policy:**

Affirmed.

**Key Questions For Authors:**

Please address the comments in the weaknesses above.

**Limitations:**

Yes, the author's have clearly situated their work in the context of robust and trustworthy learning systems in environments subject to distribution shifts and strategic manipulation.

**Strengths And Weaknesses:**

Strengths:
Controlling the tail losses in a nonstationary setting is a very relevant and interesting topic. The paper brings together several interesting ideas and provides a coherent way to tackle the problem. The numerical examples, in particular the portfolio example, are illustrative of certain properties.

Weaknesses:
Assumptions are not always clear, regret property wasn't very well motivated convincingly and the presentation of method directly as an algorithm rather compressed.
* The domain of $\lambda$ and the assumed properties of loss function $R_t(\lambda)$ are both unclear.
* RU representation of empirical CVaR_$\beta$)($R_{1:T}$) = $\min_c f(c)$ doesn't quite motivate the relevance of considering the regret of the estimator of CVaR_$\beta$)($R_{1:T}$) with respect to the auxiliary quantile parameter $c$ that is meant to track the empirical CVaR. This is not quite the *regret* of a decision-maker who is only concerned with choosing $\lambda_t$!
* There is a lot of mention of Kelly betting but the connection is not really brought forward but rather left to references such as (Orabona, 2019).

---

> ### Author Rebuttal · Authors · 2026-03-31
>
> We thank the reviewer for the careful reading and for highlighting both the relevance of online tail-risk control and the strengths of the numerical examples. We also appreciate the concerns regarding clarity of assumptions, the motivation for the regret formulation, and the presentation of the algorithm. We agree these aspects can be improved and will revise accordingly.
>
> ## Presentation and algorithm clarity
>
> We agree that the current presentation is too compressed. In the revision, we will improve the exposition by introducing a figure before the algorithm to clearly illustrate the interaction between the outer update for $\lambda_t$ and the inner update for $c_t$:
>
> - Figure: https://drive.google.com/file/d/1W91PFVy6DvmBZjlpnaLJ8d17EGIJVzOs/view?usp=sharing
>
> We will also expand the step-by-step description of the algorithm to make the flow more transparent.
>
> ## Assumptions and problem setup
>
> We will clarify the domain and assumptions as follows. The decision variable $\lambda_t$ is chosen from a compact action set $\Lambda$ (in our experiments $\Lambda = [0,1]$). Our main result only requires that the losses $R_t(\lambda)$ are uniformly bounded over $t$ and $\lambda \in \Lambda$; we do not assume convexity or smoothness in $\lambda$.
>
> The convexity used in the analysis arises only in the auxiliary RU objective with respect to the threshold variable $c$, through $f_t(c)$. We will make this distinction explicit in the problem setup and theorem statement.
>
> ## Motivation for the regret formulation
>
> We agree that this point was not sufficiently clear. The regret we introduce is not intended to represent the regret of a decision-maker whose sole control is $\lambda_t$. Rather, it is an algorithmic device induced by the Rockafellar–Uryasev representation.
>
> Specifically, the RU formulation rewrites CVaR as a minimization over an auxiliary threshold $c$. Our method jointly updates $(\lambda_t, c_t)$, where:
> - the outer update controls the average RU surrogate evaluated at $(\lambda_t, c_t)$, and
> - the inner regret guarantee ensures that the sequence $c_t$ performs nearly as well as the best fixed threshold $c^*$ in hindsight.
>
> Combining these yields control of the empirical CVaR of the realized losses $R_t(\lambda_t)$. We will revise the exposition to emphasize that $c_t$ is an auxiliary variable introduced by the variational form, while $\lambda_t$ remains the true decision variable.
>
> ## Connection to Kelly / coin-betting
>
> We agree that the connection to Kelly betting should be made more explicit. The inner one-dimensional optimization over $c_t$ is solved using a parameter-free coin-betting (Kelly-style) update. While the original work of Kelly (1956) is not adversarial, subsequent developments (e.g., Krichevsky and Trofimov, 1981; Orabona & Pal, 2016; Orabona, 2019) extend this idea to adversarial online learning.
>
> We will add a short paragraph explaining this connection and clarifying that the role of coin-betting here is to provide a parameter-free and efficient 1D no-regret update.

---

> > ### Author Rebuttal · Reviewer_Qgpv · 2026-04-02
> >
> > I think the authors have take my remarks seriously and their replies indicate they are willing to improve the paper accordingly.

---

### Official Review · Reviewer_8BAT · 2026-03-13

**Soundness:** 3
**Presentation:** 4
**Significance:** 3
**Originality:** 3
**Overall Recommendation:** 4
**Confidence:** 4

**Summary:**

The paper introduces an algorithm for controlling conditional value of risk of online sequences at pre-specified target confidence levels.
To obtain provable minimax optimal such control, they take inspiration from the conformal decision theory framework of Lekeufack et al, which connects CVaR to conformal/quantile control, and leverage two main tools: (1) the classic Rockafellar-Uryasev representation of CVaR, (2) an online coin-betting algorithm (that algorithmically implements the “Kelly criterion”). They combine these to produce the first algorithm which controls CVaR online at any target level up to a (minimax optimal) 1/sqrt(T) gap.

Additionally, to demonstrate practical utility and promise of their method, the authors conduct experiments in a couple settings in which CVaR control is appropriate and meaningful, which includes the classical area of online portfolio management as well as the emerging area of LLM toxicity mitigation. They both confirm the theoretically established bounds on CVaR control, and also showcase the inner parameter evolution of the algorithm, in stationary and adversarial environments.

**Compliance With Llm Reviewing Policy:**

Affirmed.

**Final Justification:**

The rebuttal reaffirmed my current positive opinion of the paper. Altogether, I appreciated the (I believe) modest but quite valuable contribution that this paper makes to the literature on online uncertainty quantification. The authors further addressed adding further principled empirical evaluations, and --- elsewhere --- also derived a tightening of their method (via plugging in a different type of Kelly betting). This shows the authors' commitment to further improving the manuscript as needed, and I believe that --- already in its current form but especially after the proposed edits --- the paper is well-positioned for a potential acceptance decision.

**Key Questions For Authors:**

N/A; I currently have a positive opinion of the paper in its current state.

**Limitations:**

yes

**Strengths And Weaknesses:**

To summarize upfront, I believe that this paper makes a simple but meaningful and interesting contribution to online adversarial risk control, which motivates my positive evaluation of it. In terms of soundness and presentation, I found the manuscript to be well written as well as mathematically correct and empirically sound. I will now assess the significance and originality dimensnions.

The main contribution of the paper can be quite simply summarized as noticing that the online control of CVaR, which at first glance is substantially nonlinear as it requires one to control a non-additive function of the whole sequence, can still be performed in the online-regret way --- since by leveraging the variational formulation of CVaR (a fundamental result of Rockafellar and Uryasev), CVaR can be decoupled into a time averaged objective minimized over an additional parameter. This parameter, however, can be controlled by the learner in addition to the usual iterates; and as such, an online algorithm with inner and outer-layer updates can be written down that controls a sequence of surrogates of CVaR and hence the empirical CVaR itself.

However simple, this observation that online CVaR control can be linearized by the RU reformulation, has not to my knowledge been utilized before in this context. It is also somewhat significant in informing potential future work on online UQ. In fact, as the authors aptly discuss, some works (such as GradEq of Angelopoulos et al. 2025) have recently proposed that regret minimization may not be the right approach when it comes to online uncertainty quantification, and therefore this idea of exploiting variational links to bring online UQ back into the online optimization domain goes counter to this premise in a principled way.

On the algorithmic side, once the variational equivalence has been noted as useful, the rest follows quite straightforwardly. The authors' construction utilizes Kelly betting updates, although this choice does not appear crucial to any of the conceptual takeaways --- other no regret approaches could be used in its place --- but the distinctive advantage of coin betting is that it's very performant in single-dimensional settings while not requiring any hyperparameter choice/tuning.

One could say, on a quasi-"weaknesses" side, that both this mathematical simplicity as well as the relatively stylized nature of the experiments (even in the LLM toxicity settings, one needs to somewhat artificially introduce the drift into the setting to make the evaluation more interesting/nonstationary) make this paper quite a straightforward contribution.

While it may be so, I still found the manuscript illuminating as a connecting element between several ideas that were previously separated in the literature on online risk control/UQ. Additionally, its contributions, far from being just specific to CVaR, actually point to the same template being useful for online control of a broad collection of other nonlinear risk measures that share some of the variational properties of CVaR. This justifies my overall opinion indicated above.

---

> ### Author Rebuttal · Authors · 2026-03-31
>
> Thank you for the positive and thoughtful assessment. We are glad the reviewer found the paper mathematically sound, empirically convincing, and conceptually illuminating. We especially appreciate the clear articulation of what we view as the core contribution: leveraging the Rockafellar–Uryasev variational representation to reduce online CVaR control to a regret-minimization problem over an auxiliary parameter, thereby bringing a nonlinear tail-risk objective back into the online optimization framework.
>
> We also agree that, once this variational connection is made, the resulting algorithm is relatively simple. We view this simplicity as a strength: the key novelty lies in identifying the correct reduction, which enables standard no-regret tools to be applied in a principled way. While we instantiate the method using Kelly betting, we agree this choice is not essential and will clarify its role as a parameter-free and effective 1D update.
>
> Regarding the stylized nature of the experiments, our goal was to isolate the core challenge of online CVaR control under non-stationarity in a controlled setting. To further strengthen the empirical validation, we include comparisons to a fixed-$\lambda$ baseline, which help highlight the role of adaptation.
>
> Across both LLM toxicity and portfolio settings, fixed-$\lambda$ exhibits two complementary failure modes under distribution shift:
>
> - **(i) Constraint violation in volatile regimes:**
>   In early high-volatility periods (e.g., 1996–2001), fixed-$\lambda$ exceeds the target level $\alpha=0.01$, while our method maintains valid CVaR control.
>   - Figure: https://drive.google.com/file/d/1dppX5to1nt0psE3HUd4t_SQ_Bl2yji93/view?usp=drive_link
>
> - **(ii) Over-conservatism in recovery regimes:**
>   When the environment improves (e.g., 2008–2018), fixed-$\lambda$ becomes overly conservative, whereas our method adapts upward and better utilizes the allowable risk budget.
>   - Figure: https://drive.google.com/file/d/1rEhBpIqPwb_lD_4oRbr_IKreyH5QN4_g/view?usp=drive_link
>
> These two regimes illustrate that a single fixed parameter cannot handle non-stationarity, while our method dynamically adjusts via feedback.
>
> We observe the same phenomenon in the LLM toxicity setting under adversarial sampling:
>
> - Fixed-$\lambda$ remains conservative, while our method adapts and tracks the target risk level.
>   - Figure: https://drive.google.com/file/d/1kJctuEgEDkOnY3LlU_cLx8kuOTEbXJ4t/view?usp=sharing
>
> Finally, we include an additional experiment with randomized shocks, demonstrating robustness beyond structured drift:
>
> - Our method maintains CVaR close to the target despite abrupt distribution shifts.
>   - Figure: https://drive.google.com/file/d/107AyjPxdAVPuoAXOdDahBXEIGLO0ZSmX/view?usp=sharing
>
> Together, these results highlight that adaptive online control is essential for achieving both validity and non-conservative performance under non-stationarity.

---

> > ### Author Rebuttal · Reviewer_8BAT · 2026-04-04
> >
> > Many thanks for a detailed response to my review, and I also appreciated reading the rest of the responses to other reviewers' questions and comments. I appreciate the additional experiments that the authors have provided --- comparing to the fixed-lambda variant is a natural idea and it appears to confirm the practical gains that are theoretically guaranteed by the proposed method, across several setups with different kinds of volatility. In addition, it is good to know (from the response to Reviewer Erzj) that the authors have identified that using a different variant of Kelly betting can lead to a tighter formal guarantee.
> >
> > Overall, the above instills confidence that a potential camera-ready version would include these and other helpful updates, potentially further enhancing the paper's presentation. I would like to reiterate my support for this paper, and the current positive score.

---

### Official Review · Reviewer_yhbW · 2026-03-14

**Soundness:** 3
**Presentation:** 2
**Significance:** 2
**Originality:** 2
**Overall Recommendation:** 3
**Confidence:** 2

**Summary:**

This paper studies online control of conditional value at risk (CVaR) in non stationary environment. Building on prior work of conformal risk control which assumes linearity of expectation and stationary environment, they used variational representation to overcome the non linearity of CVaR and online adaptation of auxiliary threshold using Kelly betting to overcome the non stationary environment. The paper provides a statistical guarantee on tail risk in online settings. They tested the framework in two settings, llm toxicity control under two sampling methods, and in portfolio management during extended varying market condition

**Compliance With Llm Reviewing Policy:**

Affirmed.

**Key Questions For Authors:**

while it is clear to me how expensive to have human annotation, have you explored having smaller classification LLM for machine scoring and large LLM for human scoring?

in charts 2b and 2d, why the starting point of lambda is different? I can see in the Appendix different starting points but they are different when it comes to adversarial and uniform settings, can you explain that?

**Limitations:**

the simple setup of the experiments: data is synthetic for toxic llm experiment and only two classes of portfolio management experiment.

**Strengths And Weaknesses:**

strengths:

- the idea of controlling tail risk in sequential decision setting is interesting especially in safety related challenges such as LLM toxicity.
combining conformal risk control, variational representation of CVaR, and online learning to enable CVaR control in non-stationary online settings.
- clever choice of empirical experiments domains, LLM toxicity and portfolio management clearly explains they key gaps in previous methods
- can not work on, non stationary, non linearity and online settings.

weaknesses:

-the theory part was hard to follow for readers new to the field of online adaptation, conformal calibration and risk control. the flow requires going back and forth in the section. adding more representative visuals would make the idea flows better
-In the experiment of LLM toxicity, dataset is synthetic for both the machine and human score. the score is a key part of calculating the risk.
-While the contribution is online settings, it would be interesting if it shows its performance against offline settings like having a fixed threshold. it would highlight the advantage of the proposed approach more.

---

> ### Author Rebuttal · Authors · 2026-03-31
>
> Thank you for the thoughtful review. We appreciate your positive assessment of the core idea and your recognition that controlling tail risk in sequential settings is important for applications such as LLM safety and portfolio management. We also thank you for the concrete suggestions on presentation and empirical evaluation.
>
> ## Weakness: Visual Representation
>
> We agree that the theory section can be easier to follow. In the revision, we will improve organization and include an additional figure to clarify the flow of the method:
>
> - Figure: https://drive.google.com/file/d/1W91PFVy6DvmBZjlpnaLJ8d17EGIJVzOs/view?usp=sharing
>
> ## Weakness: LLM dataset is synthetic
>
> We clarify that the LLM toxicity experiment is **semi-synthetic**. Following prior conformal tail-risk work, we use:
> - a stronger toxicity scorer as a proxy for the target (“human”) score, and
> - a biased scorer as the deploy-time machine score.
>
> This design enables a controlled benchmark where non-stationary tail behavior can be isolated and evaluated at scale.
>
> We also include an additional setting with **random shocks**, implemented via a Beta-distributed sampling scheme injected with spontaneous jumps:
>
> - Sampling regime: https://drive.google.com/file/d/1OUaQvgsRQSp1Uvd183hEF8lceF6SCvjf/view?usp=sharing
> - Results under random shocks: https://drive.google.com/file/d/107AyjPxdAVPuoAXOdDahBXEIGLO0ZSmX/view?usp=sharing
>
> In this setting, our method successfully adapts and keeps CVaR close to the target despite abrupt distribution shifts.
>
> ## Weakness: Compare to offline baseline
>
> We agree that comparison to an offline (fixed-threshold) baseline strengthens the empirical section. Our method is designed for **distribution shift**, where a fixed threshold cannot adapt to evolving tail behavior. We will make this more explicit by including a fixed-$\lambda$ baseline.
>
> In the LLM toxicity experiment, we select fixed-$\lambda$ following Chen et al. (2025). Across both adversarial and uniform settings, fixed-$\lambda$ is overly conservative, while our method adapts and keeps empirical CVaR close to the target:
>
> - **Adversarial setting:** https://drive.google.com/file/d/1kJctuEgEDkOnY3LlU_cLx8kuOTEbXJ4t/view?usp=sharing
> - **Uniform setting:** https://drive.google.com/file/d/1omXQCAB3QCtB12JrfkZFTFBfg7kNgzHw/view?usp=drive_link
>
> In portfolio management, we estimate empirical CVaR and choose the largest $\lambda$ satisfying the constraint. We observe the following:
>
> - **Early volatile periods (e.g., 1996–2001):** fixed-$\lambda$ violates the CVaR constraint (exceeds $\alpha=0.01$), while our method maintains control.
>   - Figure: https://drive.google.com/file/d/1dppX5to1nt0psE3HUd4t_SQ_Bl2yji93/view?usp=drive_link
>
> - **Crisis-first regimes (e.g., 2008–2018):** fixed-$\lambda$ is overly conservative, whereas our method adapts upward and better utilizes the risk budget.
>   - Figure: https://drive.google.com/file/d/1rEhBpIqPwb_lD_4oRbr_IKreyH5QN4_g/view?usp=drive_link
>
> - **Long horizons (1996–2025):** fixed-$\lambda$ alternates between violation and under-utilization, reflecting its inability to handle non-stationarity.
>   - Figure: https://drive.google.com/file/d/1kM9YoMNnRA6_hc7djQ7UXosNcDaCOjVp/view?usp=drive_link
>
> In contrast, our method dynamically adapts via feedback, achieving both valid CVaR control and non-conservative performance.
>
> ## Q1: Machine vs. human scoring
>
> We agree this is a natural extension. Our current semi-synthetic setup already captures this asymmetry, with a weaker deploy-time score and a stronger target score. Replacing these with a small classifier LLM for scoring and a stronger LLM-based judge or human-preference signal for the target would be a valuable next step, which we will highlight in the discussion of limitations and future work.
>
> ## Q2: Different starting points of $\lambda_t$
>
> The different apparent starting points of $\lambda_t$ in Figures 2b and 2d are due to discarding the first 100 rounds as burn-in. The first visible point is therefore not the common initialization $\lambda_1$, but the value after 100 rounds of adaptation, which differs across regimes due to different early loss sequences. We will clarify this in the caption and text.

---

### Decision · Program_Chairs · 2026-04-30

**Decision:**

Accept (regular)

**Comment:**

The paper presents a novel algorithm for maintaining the Conditional Value-at-Risk (CVaR) of online sequences at pre-specified target levels. To achieve minimax optimal control, the authors extend the conformal decision theory framework of Lekeufack et al., which links CVaR to quantile control. Their approach integrates the Rockafellar-Uryasev representation of CVaR with an online coin-betting strategy based on the Kelly criterion. This combination yields the first online algorithm capable of controlling CVaR at any target level with a minimax optimal $O(1/\sqrt{T})$ convergence gap. The authors simplify a seemingly complex problem: they show that the non-additive nature of CVaR does not preclude online control. By utilizing the variational formulation of CVaR, the paper reduces a complex sequence-level objective to a manageable online-regret framework.
All reviewers who engage in post-rebuttal discussions found this paper interesting and worth of publication. And I concur with them.
The authors are compelled to include in the camera-ready version of the paper all the promised additions/enhancements, as emerged during the discussion with the reviewers.

Note: Reviewer yhbW did not engage, hence their opinion on this paper has been downgraded.